# Glutathione synthesis in the mouse liver supports lipid abundance through NRF2 repression

Gloria Asantewaa [1,2,3], Emily T. Tuttle[2,3], Nathan P. Ward [4], Yun Pyo Kang [4], Yumi Kim [4], Madeline E. Kavanagh[5,6], Nomeda Girnius [7], Ying Chen[8], Katherine Rodriguez[2,3], Fabio Hecht [2,3], Marco Zocchi[2,3], Leonid Smorodintsev-Schiller[2,3], TashJaé Q. Scales[2,3], Kira Taylor[2,3], Fatemeh Alimohammadi[3,9], Renae P. Duncan [10,11], Zachary R. Sechrist[3,10,11], Diana Agostini-Vulaj [11], Xenia L. Schafer[1], Hayley Chang[2,3], Zachary R. Smith[2,3], Thomas N. O'Connor[2,9], Sarah Whelan[12], Laura M. Selfors [7], Jett Crowdis[7], G. Kenneth Gray [7], Roderick T. Bronson[7], Dirk Brenner [13,14,15], Alessandro Rufini[12,16], Robert T. Dirksen [9], Aram F. Hezel[2,3], Aaron R. Huber[11], Joshua Munger [1,3], Benjamin F. Cravatt [5], Vasilis Vasiliou [8], Calvin L. Cole[3,10], Gina M. DeNicola [4] & Isaac S. Harris [2,3,9] ✉

Cells rely on antioxidants to survive. The most abundant antioxidant is glutathione (GSH). The synthesis of GSH is non-redundantly controlled by the glutamate-cysteine ligase catalytic subunit (GCLC). GSH imbalance is implicated in many diseases, but the requirement for GSH in adult tissues is unclear. To interrogate this, we have developed a series of in vivo models to induce *Gclc* deletion in adult animals. We find that GSH is essential to lipid abundance in vivo. GSH levels are highest in liver tissue, which is also a hub for lipid production. While the loss of GSH does not cause liver failure, it decreases lipogenic enzyme expression, circulating triglyceride levels, and fat stores. Mechanistically, we find that GSH promotes lipid abundance by repressing NRF2, a transcription factor induced by oxidative stress. These studies identify GSH as a fulcrum in the liver's balance of redox buffering and triglyceride production.

Fundamental cellular processes, including the electron transport chain (ETC) in the mitochondria and protein folding in the ER, produce reactive oxygen species (ROS)[1]. The function of ROS can be either beneficial or damaging, depending on its cellular concentration and localization[2–4]. Antioxidative systems, including metabolites and enzymes, scavenge ROS. Disparities between antioxidant activity and ROS levels can lead to oxidative stress. Studies have shown impaired antioxidant functions connected to the onset and progression of numerous diseases[5]. Many of these studies, however, are correlative. Notably, the roles of antioxidants in normal physiology are poorly

understood, largely preventing our ability to modulate antioxidants to either prevent or treat disease.

Glutathione (GSH) is the most abundant antioxidant in the body[6] and one of the most concentrated metabolites in the cell[7]. Unlike most abundantly synthesized molecules, such as ATP or DNA, the production of GSH is controlled at the enzymatic level by a holoenzyme, glutamate-cysteine ligase (GCL), a heterodimer composed of a catalytic subunit (GCLC), and a modifier subunit (GCLM). Tissue-specific Gclc KO in mice has identified important functions for GSH in physiological settings[8–13]. However, the impact of GSH across all the tissues in adult animals is

unknown because mice that lack GCLC undergo an early embryonic lethality[14–16]. The noncatalytic modifier subunit GCLM aids GCLC in GSH synthesis. Gclm KO mice are viable[17,18], having 15–40% of normal tissue GSH levels due to GCLC expression. Using chemical inhibitors to block GCLC in animals leads to inconsistent and incomplete loss of GSH across tissues[19]. Finally, many studies have interrogated the role of requisite amino acids (i.e., cysteine, glutamate, and glycine) or metabolic cofactors that regenerate GSH (i.e., NADPH)[20–25] in GSH synthesis. Still, these molecules also have multiple GSH-independent functions, which can potentially confound interpretations. Thus, while GSH was discovered more than a century ago[26], our understanding of its physiological contributions remains incomplete.

The liver is a significant source of triglyceride production in the body[27]. It is also a primary site of drug detoxification, which relies heavily on GSH availability[28]. Indeed, GSH is reported to be produced in the liver at levels higher than any other tissue in the body[29]. Whether this GSH store influences other liver functions (i.e., de novo lipogenesis) is unclear. Mice born with liver-specific deletion of *Gclc* (using a *Gclc*[f/f] Albumin-Cre mouse strain) undergo liver failure and die shortly after birth[30]. Thus, the role of GSH synthesis in the liver and how this impacts the physiology of adult animals has yet to be fully elucidated. The development of inducible conditional knockout mouse models has revealed several aspects of tissue homeostasis in animals[31–33]. Here, we employ inducible conditional knockout mouse models to study the function of GSH in vivo. Following whole-body deletion of *Gclc*, adult mice rapidly lose weight and die, but not before showing a reduction in circulating triglyceride levels. We find that, in contrast to previous studies, liver-specific GCLC expression in adult mice is not responsible for maintaining liver survival but instead sustains the expression of lipogenic enzymes and the abundance of triglycerides in the body. Finally, we show that GSH synthesis in the liver suppresses the activity of the antioxidant transcription factor nuclear factor erythroid 2-related factor 2 (NFE2L2, also known as NRF2). By restraining NRF2 activity, GSH synthesis facilitates the expression of lipogenic enzymes and enables the supply of triglycerides into circulation. Together, our work sheds light on the core functions of GSH synthesis in vivo, where we find it necessary for sustaining lipid abundance.

## Results

### GSH synthesis is required for the survival of adult animals

To better understand the importance of GSH in adult mammals, we developed an inducible, whole-body Gclc KO mouse by breeding the *Gclc*[f/f] mouse strain[30] with the Rosa26-CreERT2 mouse strain[31]. *Gclc*[f/f] Rosa26-CreERT2 mice (hereafter referred to as Gclc KO mice) were aged to adulthood (>12 weeks old) before being given tamoxifen (i.p. injection of 160 mg/kg tamoxifen for five consecutive days, as previously described[33]) to induce Cre recombinase activity and *Gclc* excision (Fig. 1A). *Gclc*[f/f] mice (hereafter described as Gclc WT) were also treated with tamoxifen to control for potential confounding effects associated with tamoxifen treatment. Approximately 12–15 days following tamoxifen treatment, mouse tissues were analyzed. A robust decrease in *Gclc* mRNA and GCLC protein (Fig. 1B, C) and lower GSH levels (Fig. 1D) were observed across most tissues in Gclc KO mice compared with Gclc WT mice. Particularly, the induced

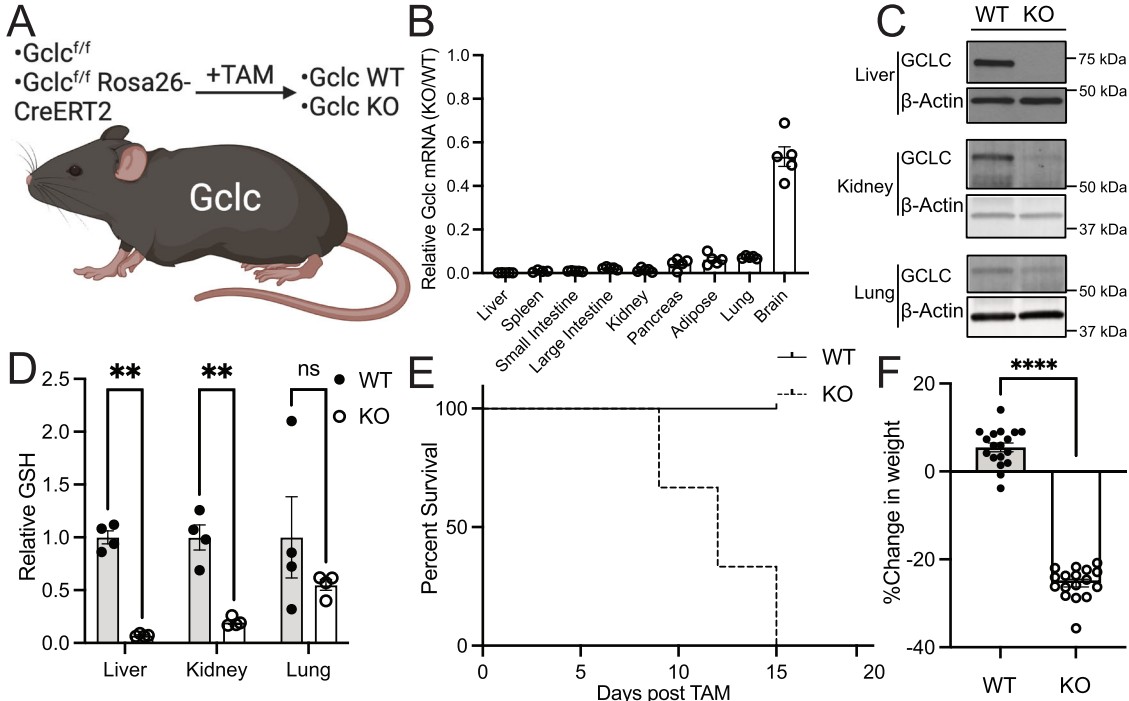

**Fig. 1 | GSH synthesis is required for the survival of adult animals. A** Schematic of the whole-body inducible *Gclc* knockout mouse model. Knockout of the whole-body *Gclc* was induced by a 5-day daily 160 mg/kg tamoxifen administration. Mice were sacrificed 12–15 days following tamoxifen administration. **B** Relative expression of *Gclc* mRNA in tissues from the KO (*n* = 4 (spleen); *n* = 5 (all other annotated tissues)) compared to WT (*n* = 5) mice 12–15 days following tamoxifen administration. Expression levels were normalized to the expression of the reference gene *Rps9*. **C** Representative immunoblot analysis of GCLC protein in the liver, kidney, and lung of WT and KO mice 12–15 days following tamoxifen administration. Data shown are representative of at least 3 replicates. **D** Relative GSH abundance in the liver, kidney, and lungs of WT (*n* = 4) and KO (*n* = 4) mice 12–15 days following tamoxifen administration. A two-way ANOVA with subsequent Šidák's multiple comparisons test was used to determine statistical significance (WT vs. KO: Liver *P* value = 0.0031, Kidney *P* value = 0.0099, Lung *P* value = 0.2054). **E** Percent survival of WT (*n* = 9) and KO (*n* = 11) mice following tamoxifen treatment. Loss of greater than 20% body weight resulted in a humane endpoint for mice. **F** Percent change in body weight in WT (*n* = 18) and KO (*n* = 17) mice at death. Initial weight measurements were collected on Day 0 of tamoxifen administration, and final weight measurements were collected at endpoints. An unpaired two-tailed *t* test was used to determine statistical significance (WT vs. KO *P* value < 0.0001). Indicated *n* values represent biologically independent samples from mice. ns = not significant, *P* value < 0.05, **P* value < 0.01, ****P* value < 0.0001. (A) created with BioRender.com, released under a Creative Commons Attribution-NonCommercial-NoDerivs 4.0 International license.

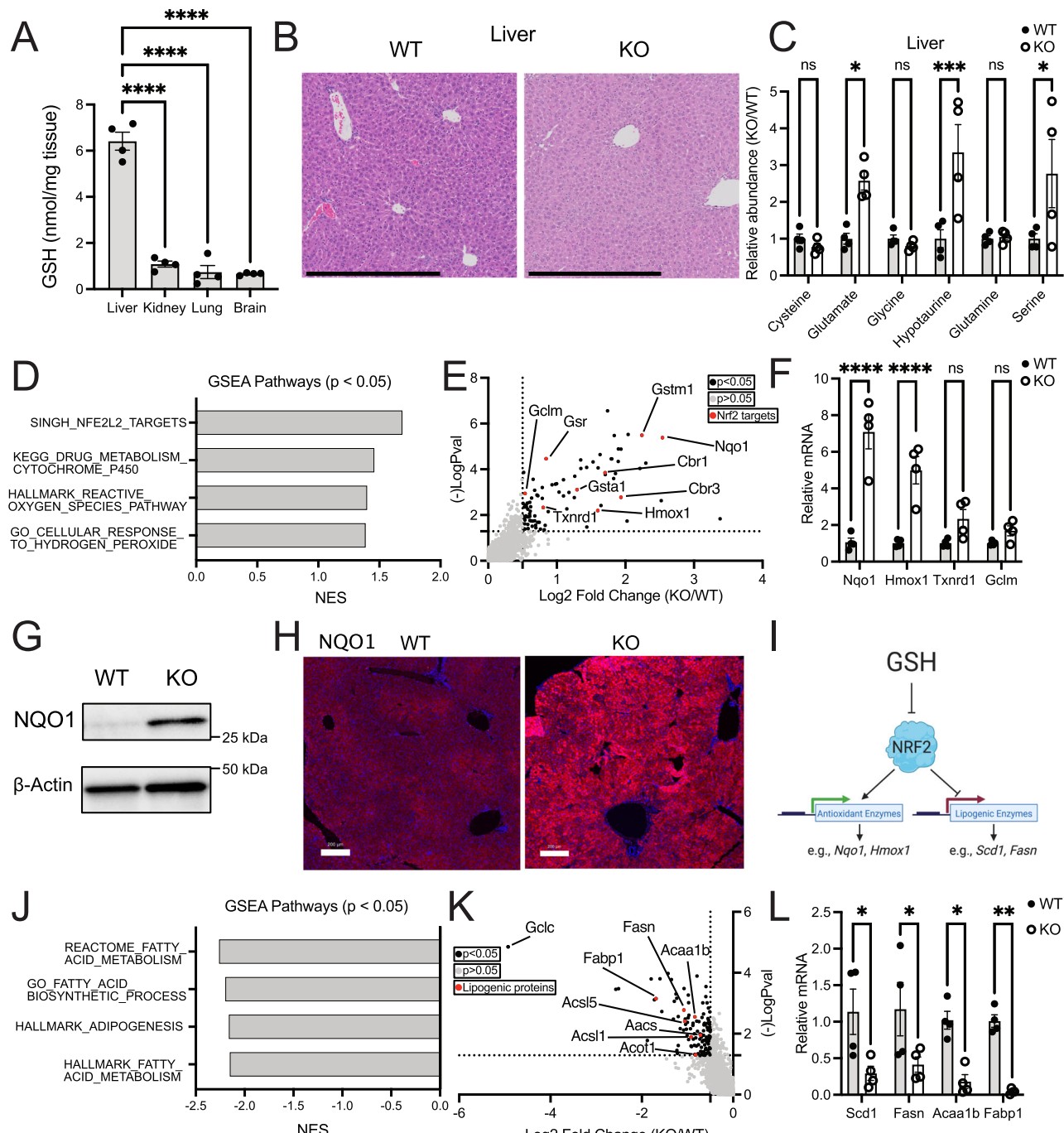

depletion of GSH in organs of adult animals was greater than previously observed depletions using pharmacologic approaches[19]. Notably, the brain had the lowest reduction in *Gclc* mRNA abundance (Fig. 1B). Although tamoxifen and its metabolites (i.e., 4-hydroxytamoxifen) can cross the blood-brain barrier, these compounds have reduced bioavailability in the brain compared to other tissues[34]; thus, potentially accounting for the lack of effective deletion observed. Deletion of *Gclc* caused a dramatic weight loss in mice, resulting in humane endpoints (Fig. 1E, F). A maximal deletion of *Gclc* was required to cause weight loss and reduction in survival in mice, as heterozygous deletion did not cause mice to lose weight (Fig. S1A–S1C), suggesting that even minimal production of GSH can maintain homeostatic processes in the body. These phenotypes were not attributed to tamoxifen treatment, as treatment of control mice (*Gclc*^f/w mice) with tamoxifen failed to cause any weight changes (Fig. S1D). Further, no difference

between biological sexes was observed in Gclc KO mice (Fig. S1E). Interestingly, the weight loss in Gclc KO mice was not associated with a corresponding loss in appetite, as no change in food consumption was observed (Fig. S1F). These findings suggest that GSH is required to maintain survival in adult mice.

## Loss of GSH synthesis induces transcription of NRF2 target genes and repression of lipogenic genes

GSH synthesis is required for numerous enzymatic processes across a range of cell types in the body. It is also a co-factor for GPX4-mediated lipid detoxification[35], and GPX4 activity is necessary for the survival of several tissues, including the liver[36] and kidney[37]. Further, the liver is reported to be one of the largest producers of GSH[29]. In line with previous findings, we show that the liver contained the highest levels of GSH compared to other organs (Fig. 2A). Surprisingly, Gclc KO mice

**Fig. 2 | Liver tissue from Gclc KO mice has induced NRF2 target genes and repressed lipogenic gene expression. A** GSH quantity (normalized to mg tissue weight) in the liver, kidney, lung, and brain from Gclc WT mice ($n = 4$). A one-way ANOVA with subsequent Dunnett's multiple comparisons test was used to determine statistical significance (Liver vs. Kidney $P$ value < 0.0001, Liver vs. Lung $P$ value < 0.0001, Liver vs. Brain $P$ value < 0.0001). **B** Representative H&E-stained liver slides in WT and KO mice. Scale bars = 500 μm. Data shown is representative of at least three replicates. **C** Relative abundance of GSH precursors and their related metabolites in the serum of WT ($n = 4$) and KO ($n = 4$) mice. A two-way ANOVA with subsequent Šidák's multiple comparisons test was used to determine statistical significance (WT vs. KO: Cysteine $P$ value = 0.9987, Glutamate $P$ value = 0.0278, Glycine $P$ value = 0.9993, Hypotaurine $P$ value = 0.0004, Glutamine P value > 0.9999, Serine P value = 0.0105). **D** Gene Set Enrichment Analysis (GSEA) of oxidative stress-related pathways in the liver of KO ($n = 4$) compared to WT ($n = 4$) mice. Enrichment p-values were calculated using an adaptive multi-level split Monte Carlo scheme and were corrected for multiple testing using Benjamini and Hochberg false discovery rate. **E** Proteomic analysis of upregulated liver proteins in KO ($n = 6$) compared to WT ($n = 6$) mice. Black data points = proteins with $P$ value < 0.05 and log2 fold change >1. Red data points = annotated NRF2 target proteins. **F** Relative mRNA expression of annotated NRF2 target genes in the liver of WT ($n = 4$) and KO ($n = 4$) mice. Expression levels were normalized to the expression of the reference gene *Rps9*. A two-way ANOVA with subsequent Šidák's multiple comparisons test was used to determine statistical significance (WT vs. KO: *Nqo1* P

value < 0.0001, *Hmox1* P value < 0.0001, *Txnrd1* P value = 0.2255, *Gclm* P value = 0.7934). **G** Representative immunoblot analysis of NQO1 in the liver of WT and KO mice. Data shown is representative of at least three replicates. **H** Representative immunofluorescence images of the liver from WT and KO mice stained with an antibody against NQO1. Scale bars = 200 μm. Data shown is representative of at least three replicates. **I** Schematic of the proposed mechanism of NRF2-dependent repression of lipogenic gene expression. **J** GSEA of lipogenic-related pathways in the liver of KO ($n = 4$) compared to WT ($n = 4$) mice. Enrichment $p$ values were calculated using an adaptive multi-level split Monte Carlo scheme and were corrected for multiple testing using Benjamini and Hochberg false discovery rate. **K** Proteomic analysis of downregulated liver proteins in KO ($n = 6$) compared to WT ($n = 6$) mice. Black data points = proteins with $P$ value < 0.05 and log2 fold change > 1. Red data points = annotated lipogenic proteins. **L** Relative mRNA expression of annotated lipogenic genes in the liver of WT ($n = 4$) and KO ($n = 4$) mice. Expression levels were normalized to the expression of the reference gene *Rps9*. A two-way ANOVA with subsequent Šidák's multiple comparisons test was used to determine statistical significance (WT vs. KO: *Scd1 P* value = 0.0160, *Fasn P* value = 0.0336, *Acaa1b* P value = 0.0162, *Fabp1* P value = 0.0051). Indicated $n$ values represent biologically independent samples from mice. Data are shown as mean ± SEM. ns = not significant, *$P$ value < 0.05, **$P$ value < 0.01, ****$P$ value < 0.0001. (I) created with BioRender.com, released under a Creative Commons Attribution-NonCommercial-NoDerivs 4.0 International license.

showed no signs of liver damage, both by histology and serum biochemistry markers (Fig. 2B and S2A). While liver damage markers were increased in the serum of Gclc KO mice, their levels fell within the physiological range reported in healthy mice[38]. Indeed, other tissues from the Gclc KO mice, including the kidney, pancreas, and spleen, were also histologically similar to tissues from the Gclc WT mice (Fig. S2B–S2D). This suggests that, compared to the requirement for GSH synthesis in embryonic development[14–16], GSH synthesis is dispensable to several organs in adult animals.

The amino acids required to synthesize the GSH (cysteine, glutamate, and glycine) are also essential to many cellular processes, ranging from energy generation to protein synthesis[39]. Further, these precursors can be converted into other metabolites. Cysteine can be metabolized into cysteine sulphinate (via CDO1), which then can produce hypotaurine (via CSAD)[40]. Glycine can be interchanged with serine[41], and glutamate can be converted to glutamine[42]. To determine if GSH precursors accumulate and are possibly rerouted upon halting GSH synthesis in vivo, we performed targeted metabolite analysis on tissues from Gclc WT and KO mice. In liver tissue from Gclc KO mice, we observed an accumulation of glutamate but no difference in cysteine and glycine levels (Fig. 2C). We hypothesized that the lack of accumulated cysteine and glycine could be due to these metabolites entering alternative metabolic pathways. Indeed, we found hypotaurine and serine levels increased in the liver of Gclc KO mice (Fig. 2C). These data suggest that in the absence of GSH synthesis, the liver accumulates GSH precursors and can re-route them into alternative metabolic pathways.

We hypothesized that, in addition to metabolic alterations, other changes were occurring in the liver of Gclc KO mice, which could explain the lack of damage in the tissue. KEAP1 is oxidized and inactivated upon oxidative stress, stabilizing the antioxidant transcription factor NRF2[43]. Hepatocytes induce *Nrf2* and reprogram their antioxidant pathways to deal with oxidative stress[44,45]. RNA-seq and subsequent gene set enrichment analysis (GSEA) revealed that the liver tissue from Gclc KO mice was enriched in several gene signatures associated with stress responses, including those related to the activation of NRF2 (Fig. 2D and Supplementary Data S1). Consistent with the transcriptomic findings, proteomic analysis of liver from Gclc KO mice revealed that most significantly elevated proteins in Gclc KO liver were encoded by NRF2 target genes, including the canonical NRF2 target NQO1 (NAD(P)H Quinone Dehydrogenase 1) (Fig. 2E–H and Supplementary Data S2). This stress response was not limited to the

liver but instead observed across multiple tissues from Gclc KO mice, including the kidney and colon (Fig. S3A). These results suggest that upon depletion of GSH, tissues undergo reprogramming to hyperactivate NRF2, possibly as a mechanism to upregulate adjacent antioxidant pathways and buffer oxidative stress induced upon GSH depletion.

Activation of NRF2 is reported to coincide with the repression of enzymes involved in adipogenesis (Fig. 2I)[46,47]. In addition to increased transcription of NRF2 target genes, livers from Gclc KO mice had lower expression of lipogenic genes and proteins (Fig. 2J–L, S3B, and Supplementary Data S1), such as stearoyl-Coenzyme A desaturase 1 (SCD1), a key enzyme in adipogenesis. The co-occurrence of high NRF2 target genes with low lipogenic genes was shared by some tissues but not all (Fig. S3C). These results indicate that following GSH depletion, lipogenic gene expression is repressed. The extent to which this impacted lipogenic products (i.e., fatty acids and triglycerides) and whether this was connected to NRF2 activation was unclear.

## GSH synthesis is required to maintain circulating triglycerides

Systemic depletion of GSH disproportionally impacted body weight. Multiple tissues showed lower lipogenic enzyme expression, suggesting lipid production might also be impaired. To examine this, we performed lipidomics on several tissues from Gclc KO mice (Figs. 3A, B, S4A, B, and Supplementary Data S3). We found that triglycerides were among the most significantly depleted lipid species across tissues from Gclc KO mice. Since this appeared to be a global trend, we hypothesized that circulating triglycerides were depleted in Gclc KO mice. Indeed, lipidomics revealed that nearly all the significantly depleted lipid species in serum from Gclc KO mice were triglycerides (Fig. 3C and Supplementary Data S3). This phenotype was confirmed by targeted measurement of triglycerides in serum (Fig. 3D). Together, these data demonstrate that the abundance of triglycerides in Gclc KO mice is lower, especially in circulating pools.

Next, we examined whether supplementing with a diet rich in lipids could reverse these reduced triglyceride and weight loss phenotypes. Supplying Gclc KO mice with a high-fat diet (HFD) rescued the decreased triglyceride levels in the serum (Fig. 3E). However, Gclc KO mice still lost substantial weight while on an HFD, although to a lesser extent than normal chow (Fig. 3F). These data suggest that maintenance of circulating triglycerides and sustaining body weight were independent phenotypes in Gclc KO mice.

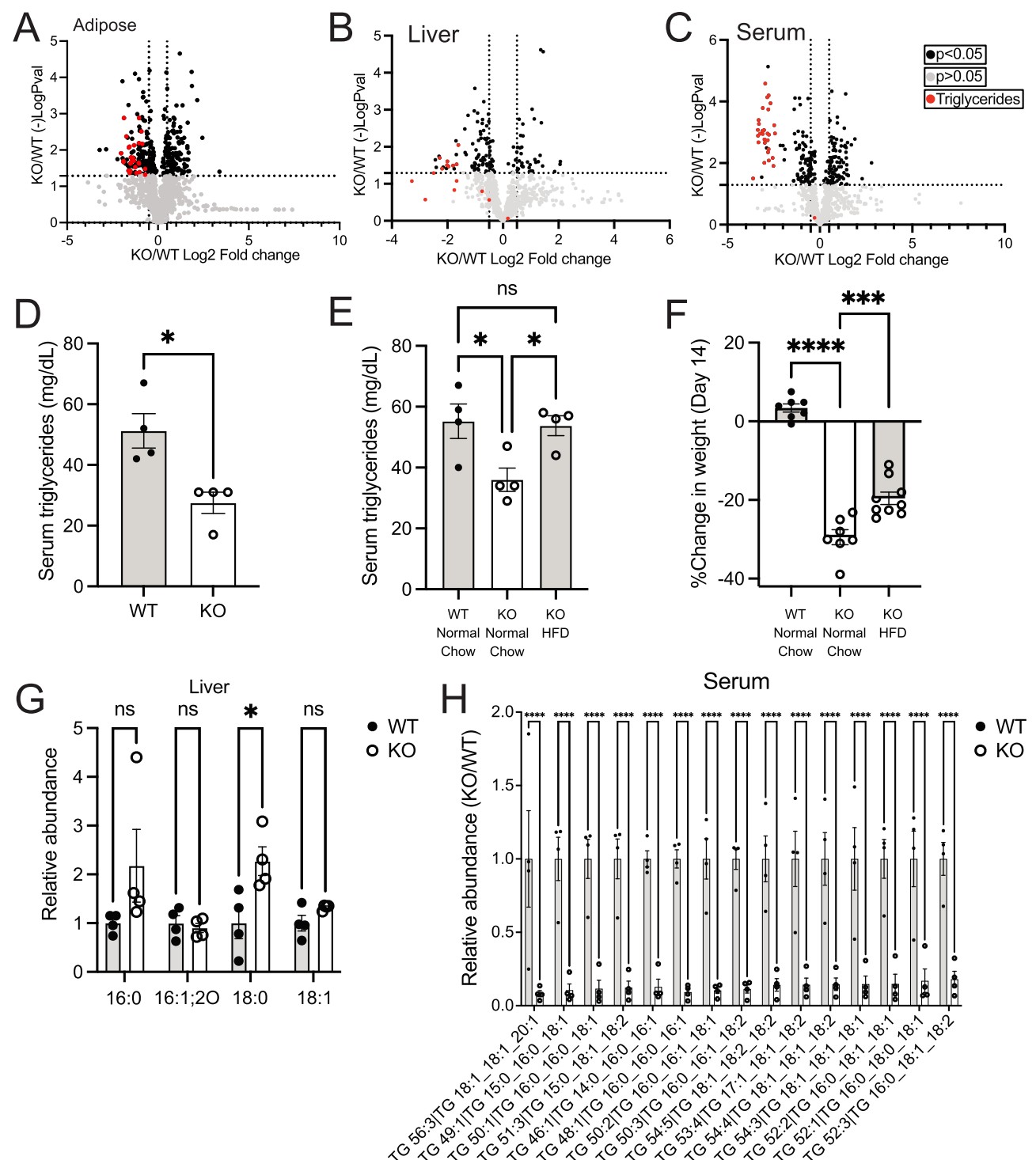

Loss of GSH synthesis resulted in a decreased expression of lipogenic enzymes, including SCD1. SCD1 catalyzes the desaturation of palmitic acid (16:0) into palmitoleic acid (16:1) and stearic acid (18:0) into oleic acid (18:1)[48]. Thus, we measured the abundance of these species in the liver of Gclc KO mice. Levels of SCD1 substrates (i.e., palmitic acid and stearic acid) were elevated in the liver of Gclc KO mice (Fig. 3G). Further, the serum and liver from Gclc KO mice had decreased levels of triglycerides that contained palmitoleic acid and oleic acid (Fig. 3H and S4C). Together, these data demonstrate that ablation of GSH synthesis in mice results not only in reduced expression of lipogenic enzymes but also lower levels of their corresponding lipid species.

**Liver-specific GSH synthesis promotes lipid abundance**

Whole-body ablation of GSH synthesis impacted lipid abundance in vivo. Synthesis of GSH and de novo lipids occurs in the liver. Whether liver-specific GSH synthesis was responsible for downregulated lipogenic gene expression and reduced serum triglycerides or whether these phenotypes were a byproduct of a liver-independent process in Gclc KO mice was unclear. Notably, Gclc KO mice rapidly lose weight and reach humane endpoints early (Fig. 1E, F), making it difficult to extricate the other phenotypes potentially caused by the loss of GSH. To interrogate this, we utilized an AAV-TBG-Cre viral delivery system, which produces liver-specific Cre expression and subsequent genetic deletion[49,50]. $Gclc^{w/f}$ or $Gclc^{w/w}$ mice (Gclc WT) and $Gclc^{f/f}$ mice (hereafter

**Fig. 3 | The abundance of triglycerides is lower following GSH depletion in mice.** Lipidomic analysis of (**A**) adipose tissue, (**B**) liver tissue, and (**C**) serum in KO ($n = 4$) compared to WT ($n = 4$) mice. Black data points = lipid species with a $p < 0.05$ and log2 fold change >1. Red data points = triglycerides. **D** Triglyceride levels in the serum of WT ($n = 4$) and KO ($n = 4$) mice. An unpaired two-tailed t-test was used to determine statistical significance (WT vs. KO P value = 0.0119). **E** Triglyceride levels in the serum for WT ($n = 4$) and KO ($n = 4$) mice fed with either normal chow or a high-fat diet (HFD) for 14 days post-treatment with tamoxifen. A one-way ANOVA with subsequent Tukey's multiple comparisons test was used to determine statistical significance (WT Normal Chow vs. KO Normal Chow P value = 0.0307, WT Normal Chow vs. KO HFD P value = 0.9683, KO Normal Chow vs. KO HFD P value = 0.0447). **F** Percent change in body weight of WT ($n = 7$) and KO mice fed with either normal chow ($n = 7$) or a high-fat diet (HFD; $n = 9$) 14 days post-treatment with tamoxifen. A one-way ANOVA with subsequent Tukey's multiple comparisons test was used to determine statistical significance (WT Normal Chow vs. KO Normal Chow P value < 0.0001, WT Normal Chow vs. KO HFD P value =<0.0001, KO Normal Chow vs. KO HFD P value = 0.0006). **G** Relative abundance of select SCD1 fatty acids products in the liver from WT ($n = 4$) and KO ($n = 4$) mice. 16:0 = palmitic acid; 16:1;2 O= oxidized palmitoleic acid; 18:0 = stearic acid; 18:1 = oleic acid. A two-way ANOVA with subsequent Šidák's multiple comparisons test was used to determine statistical significance (WT vs. KO: 16:0 P value = 0.0603, 16:1;2 O P value = 0.9991, 18:0 P value = 0.0377, 18:1 P value = 0.9268). **H** Relative abundance of select triglyceride species in the serum from WT ($n = 4$) and KO ($n = 4$) mice. A two-way ANOVA with subsequent Šidák's multiple comparisons test was used to determine statistical significance (WT vs. KO for all comparisons indicated: **** P value < 0.0001). Indicated $n$ values represent biologically independent samples from mice. Data are shown as mean ± SEM. *ns* not significant, * P value < 0.05, ** P value < 0.01, **** P value < 0.0001.

referred to as Gclc L-KO) were monitored following tail-vein injection with AAV-TBG-Cre (Fig. 4A). The liver from Gclc L-KO mice showed rapid depletion of *Gclc* mRNA and progressive loss of GCLC protein and GSH levels (Fig. 4B, C). No differences in *Gclc* mRNA were observed in surrounding tissues (Fig. S5A), suggesting that the deletion of *Gclc* was localized to the liver. Further, no differences in GSH levels were found in surrounding issues (i.e., the kidney) (Fig. S5B), suggesting that catabolism liver-derived circulating GSH was not contributing to GSH levels in tissues. Unlike whole-body Gclc KO mice, liver-specific Gclc L-KO mice did not have any difference in survival or body weight following *Gclc* deletion (Fig. S5C, D). Gclc L-KO mice showed no overt histological signs of liver damage (Fig. S5E), although analysis of hepatocellular injury and serum AST and ALT levels showed a significant increase compared to Gclc WT mice (Fig. S5F-S5H). Further histopathological analyses showed that the elevation in liver damage biomarkers observed in the Gclc L-KO mice was not associated with increased markers of apoptosis but was associated with increased markers of necroptosis (Fig. S6A). Notably, no significant increases in markers of inflammation or oxidative stress were observed in the liver of Gclc L-KO mice compared to Gclc WT mice (Fig. S6B). These data demonstrate that deletion of the GCLC protein can be induced in the liver of adult animals. Importantly, these phenotypes contrast with the non-inducible liver-specific Gclc KO mouse strain (*Gclc*^f/f Albumin-Cre), which develops liver failure and dies shortly after birth[30].

Next, we examined whether livers from liver-specific Gclc KO mice induced an antioxidant response and had alterations in lipids and lipid-generating enzymes as initially observed in whole-body Gclc KO mice. Liver tissue from Gclc L-KO mice showed enrichment of NRF2 protein and downstream target genes (Fig. 4D–F, S7A, B, and Supplementary Data S4). Like the whole-body Gclc KO mice, we observed a decrease in lipogenic factors and serum triglycerides over time (Fig. 4G–I, S7C, and Supplementary Data S4). This was associated with changes in serum glucose levels but not serum cholesterol levels (Fig. S7D-S7E). Additionally, the epididymal white adipose tissue, one of the major triglyceride storage tissues, was found not to be impacted by the loss of *Gclc* and reduced triglycerides in the Gclc L-KO mice (Fig. S7F). Further, we found this decreased serum triglyceride levels to be reversed when mice were placed on an HFD (Fig. 4J). Interestingly, metabolite analysis revealed decreased expression of acetyl-CoA, an essential precursor for lipogenesis, and downstream acetyl-CoA metabolites (Fig. S7G). These data demonstrate that GSH synthesis in the liver is responsible for maintaining lipogenic gene expression and sustaining circulating triglyceride levels in vivo.

Our findings showed that liver-specific *Gclc* deletion results in increased Nrf2 activity, impaired expression of lipogenic enzymes, and decreased circulating triglycerides. However, the extent of GSH depletion required to impact triglyceride metabolism was unknown. To examine this, we treated mice with titrated amounts of the AAV-TBG-Cre virus to induce varying degrees of *Gclc* deletion. Higher viral concentration resulted in greater induction of NRF2 activity (Fig. S8A-S8B). This was associated with a concurrent reduction in lipogenic gene expression and serum triglycerides (Fig. S8C-S8E). Particularly, we found the repression of lipogenic genes required a strong induction of NRF2 activity, as mild induction of NRF2 activity ($0.5 \times 10^{11}$ pfu) was not sufficient to repress lipogenic gene expression. Further, we observed higher viral concentration correlated with increased expression of serum markers of liver damage (Fig. S8F-S8G). Together, these findings suggest that even minimal levels of GSH in the liver can maintain its ability to support serum triglyceride levels.

GCLC catalyzes the rate-limiting step of GSH synthesis. However, whether its impact on lipid-related phenotypes is due to non-canonical activities of GCLC was unclear. Indeed, GCLC has non-canonical functions in sequestering glutamate levels to prevent ferroptosis[51]. To examine the extent to which the phenotypes observed were specifically caused by lower GSH levels, we rescued GSH depletion in liver-specific Gclc KO mice with GSH mono ethyl ester (GSH-ee). Treatment of Gclc L-KO with GSH-ee reduced the observed increased Nrf2 target gene expression to basal levels in a dose-dependent manner (Fig. 4K and S9A). However, this was not sufficient to rescue the decreased lipogenic factors or decreased circulating triglycerides (Fig. 4L, M and S9B), potentially due to the technical limitations associated with delivering GSH to adult animals. These data indicate that GSH depletion is directly responsible for the observed induction of NRF2 activity in Gclc L-KO mice. However, while technical limitations may exist regarding adequate GSH rescue, we cannot rule out non-canonical and non-enzymatic functions of GCLC contributing to lipid-related phenotypes.

## GSH supports lipid abundance by repressing NRF2 activation

GSH is reported to contribute to lipid homeostasis in the body through several mechanisms[13,52–56]. However, our data suggest that following GSH depletion, NRF2 is stabilized, and this stabilization is associated with decreased expression of lipogenic enzymes and lower serum triglyceride levels. To test the involvement of NRF2, we bred the *Gclc*^f/f mouse strain with the *Nrf2*^f/f mouse strain[57,58] and induced a liver-specific double deletion with the AAV-TBG-Cre virus (hereafter referred to as L-DKO) (Fig. 5A). *Gclc* and *Nrf2* mRNA levels were decreased in L-DKO mice three weeks following injection of the AAV-TBG-Cre virus (Fig. 5B, C), although the *Nrf2* gene was less efficiently deleted compared to the deletion of *Gclc*. Nonetheless, the deletion of *Nrf2* prevented the increased expression of NRF2 protein and the induction of NRF2 targets upon *Gclc* deletion in liver tissue (Fig. 5D, E). Notably, the repression of lipogenic gene and protein expression and the decreased serum triglyceride levels seen in Gclc L-KO mice were reversed in L-DKO mice (Fig. 5F–J, and Supplementary Data S5). For certain lipogenic genes, however, the reversal of downregulated expression was not complete (Fig. 5F), suggesting the potential involvement of NRF2-independent pathways. Together, these results indicate that GSH supports lipid abundance by preventing NRF2 activation in the liver.

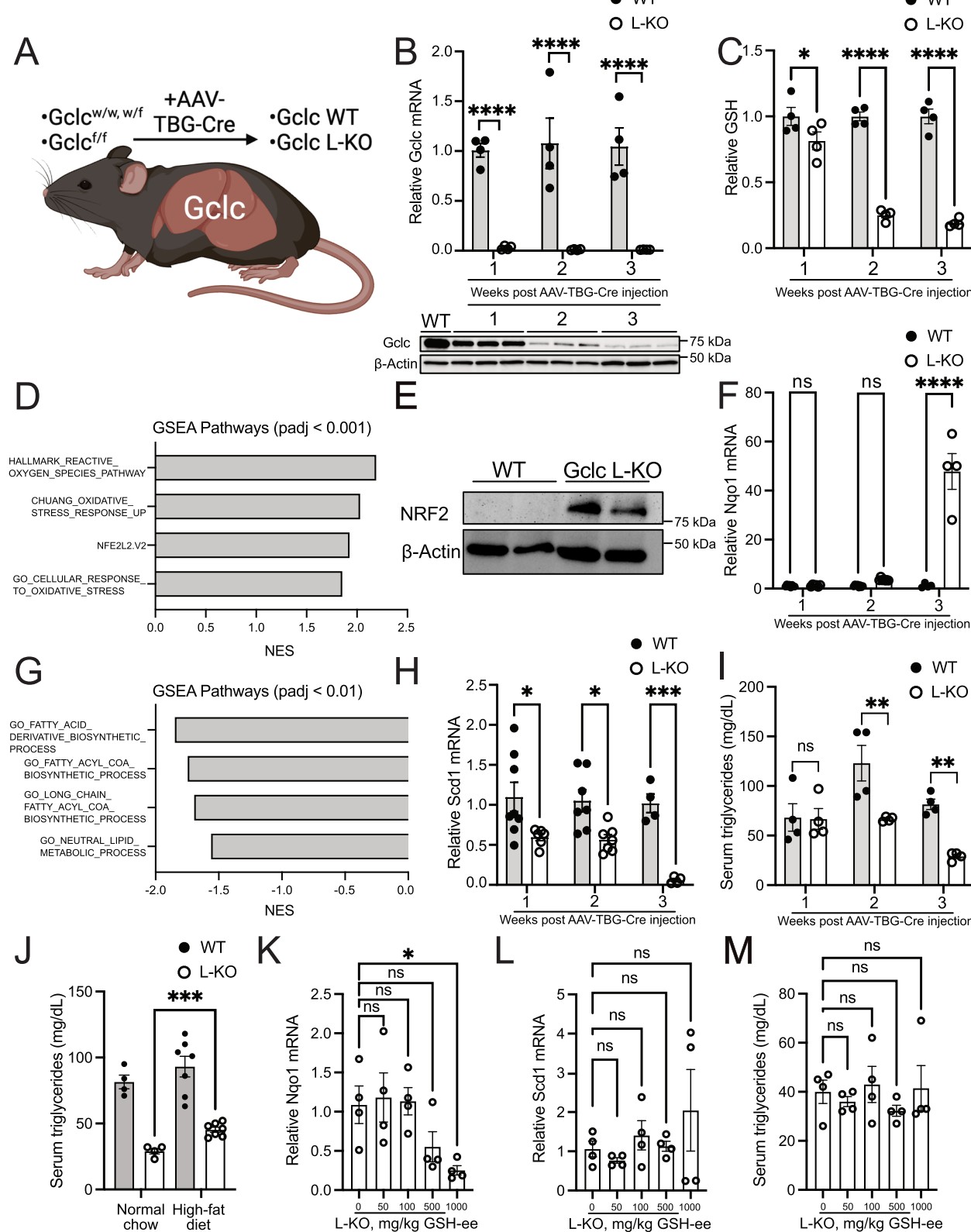

Activation of NRF2 prevents tissue damage under pathological conditions[59]. Since we observed an activation of NRF2 upon GSH depletion, we expected NRF2 to be essential for preventing liver damage. Surprisingly, L-DKO mice did not die or significantly lose weight shortly after genetic deletion was induced (Fig. S10A). Further, L-DKO mice did not have significantly increased levels of serum liver damage markers (Fig. S10B-S10C) nor showed signs of inflammation, fibrosis, or liver failure (Fig. S10D and S11). These results demonstrate that the expression of NRF2 in the liver is acutely dispensable under GSH-depleted conditions. Further, this data suggests that rather than buffering oxidative stress to prevent liver damage, GSH synthesis is crucial for maintaining low NRF2 levels and sustaining lipid production by the liver.

NRF1 is a transcription factor that belongs to the same transcription factor family as NRF2. While NRF1 and NRF2 are differentially regulated

**Fig. 4 | Liver-specific GCLC expression sustains lipid synthesis and represses NRF2 activation. A** Schematic of inducible liver-specific *Gclc* deletion (L-KO). Liver-specific Gclc knockout was induced using the AAV-TBG-Cre. All mice were sacrificed 21 days post AAV-TBG-Cre injection unless otherwise indicated. **B** (Top) Relative abundance of *Gclc* mRNA in the liver of WT ($n = 4$) and L-KO ($n = 4$) mice, 1–3 weeks post-treatment with AAV-TBG-Cre. Expression levels of mRNA were normalized to the expression of the reference gene *Rps9*. A two-way ANOVA with subsequent Šidák's multiple comparisons test was used to determine statistical significance (WT vs. L-KO: 1 week *P* value = 0.0002, 2 weeks *P* value < 0.0001, 3 weeks *P* value < 0.0001). (Bottom) Immunoblot analysis of GCLC protein in the liver of WT and L-KO mice 1–3 weeks post-treatment with AAV-TBG-Cre. **C** Relative abundance of GSH in liver tissue of WT ($n = 4$) and L-KO ($n = 4$) mice in 1–3 weeks following AAV-TBG-Cre treatment. A two-way ANOVA with subsequent Šidák's multiple comparisons test was used to determine statistical significance (WT vs. L-KO: 1 week *P* value = 0.0424, 2 weeks *P* value < 0.0001, 3 weeks *P* value < 0.0001). **D** GSEA of oxidative stress-related pathways in the liver of L-KO ($n = 4$) compared to WT ($n = 4$) mice following treatment with AAV-TBG-Cre. Enrichment *p* values were calculated using an adaptive multi-level split Monte Carlo scheme and were corrected for multiple testing using Benjamini and Hochberg false discovery rate. **E** Representative immunoblot analysis of NRF2 in the liver of WT and L-KO mice following treatment with AAV-TBG-Cre. Data shown are representative of at least three replicates. **F** Relative mRNA levels of *Nqo1* in WT ($n = 8$ for 1 and 2-week timepoints and $n = 4$ for 3-week timepoint) and L-KO ($n = 6$ for 1-week timepoint, $n = 7$ for 2-week timepoint and $n = 4$ for 3-week timepoint) mice,1–3 weeks following treatment with AAV-TBG-Cre. Expression levels were normalized to the expression of the reference gene *Rps9*. A two-way ANOVA with subsequent Šidák's multiple comparisons test was used to determine statistical significance (WT vs. L-KO: 1 week *P* value = 0.9998, 2 weeks *P* value = 0.6378, 3 weeks *P* value < 0.0001). **G** GSEA of lipogenic-related pathways in the liver of KO ($n = 4$) compared to WT ($n = 4$) mice following treatment with AAV-TBG-Cre. Enrichment *p* values were calculated using an adaptive multi-level split Monte Carlo scheme and were corrected for multiple testing using Benjamini and Hochberg false discovery rate. **H** Relative mRNA levels of *Scd1* in WT ($n = 8$ for 1 and 2-week timepoints and $n = 4$ for 3-week timepoint) and L-KO ($n = 6$ for 1-week timepoint, $n = 7$ for 2-week

timepoint and $n = 4$ for 3-week timepoint) mice, 1–3 weeks following treatment with AAV-TBG-Cre. Expression levels were normalized to the expression of the reference gene *Rps9*. A two-way ANOVA with subsequent Šidák's multiple comparisons test was used to determine statistical significance (WT vs. L-KO: 1 week *P* value = 0.0192, 2 weeks *P* value = 0.0222, 3 weeks *P* value = 0.0005). **I** Triglyceride levels in the serum of WT ($n = 4$) and L-KO ($n = 4$) mice in 1–3 weeks following treatment with AAV-TBG-Cre. A two-way ANOVA with subsequent Šidák's multiple comparisons test was used to determine statistical significance (WT vs. L-KO: 1 week *P* value = 0.9995, 2 weeks *P* value = 0.0038, 3 weeks *P* value = 0.0069). **J** Triglyceride levels in the serum of L-KO mice fed normal chow ($n = 4$) and HFD ($n = 7$) 3 weeks following treatment with AAV-TBG-Cre. An unpaired two-tailed t-test was used to determine statistical significance (L-KO Normal Chow vs. L-KO High-fat Diet *P* value = 0.0004). Relative mRNA levels of (**K**) *Nqo1* and (**L**) *Scd1* in Gclc L-KO mice ($n = 4$) following treatment with GSH-ee (0, 50, 100, 500, and 1000 mg/kg) from days 17–20 post-AAV-TBG-Cre injection. mRNA levels were normalized to the expression of the reference gene *Rps9*. A one-way ANOVA with subsequent Dunnett's multiple comparisons test was used to determine statistical significance (**K**): 0 mg/kg GSH-ee vs. 50 mg/kg GSH-ee *P* value = 0.9933, 0 mg/kg GSH-ee vs. 100 mg/kg GSH-ee *P* value = 0.9996, 0 mg/kg GSH-ee vs. 500 mg/kg GSH-ee *P* value = 0.2648, 0 mg/kg GSH-ee vs. 1000 mg/kg GSH-ee *P* value = 0.0454. **L**: 0 mg/kg GSH-ee vs. 50 mg/kg GSH-ee *P* value = 0.9816, 0 mg/kg GSH-ee vs. 100 mg/kg GSH-ee P value = 0.9664, 0 mg/kg GSH-ee vs. 500 mg/kg GSH-ee *P* value > 0.999, 0 mg/kg GSH-ee vs. 1000 mg/kg GSH-ee *P* value = 0.4731. **M** Triglyceride levels in the serum of Gclc L-KO mice ($n = 4$) following treatment with GSH-ee (0, 50, 100, 500, and 1000 mg/kg) from days 17–20 post-AAV-TBG-Cre injection. A one-way ANOVA with subsequent Dunnett's multiple comparisons test was used to determine statistical significance (0 mg/kg GSH-ee vs. 50 mg/kg GSH-ee *P* value = 0.9663, 0 mg/kg GSH-ee vs. 100 mg/kg GSH-ee *P* value = 0.9879, 0 mg/kg GSH-ee vs. 500 mg/kg GSH-ee *P* value = 0.7552, 0 mg/kg GSH-ee vs. 1000 mg/kg GSH-ee *P* value = 0.9991). Indicated *n* values represent biologically independent samples from mice. Data are shown as mean ± SEM. ns = not significant, **P* value < 0.05, ***P* value < 0.01, ****P* value < 0.001, *****P* value < 0.0001. (A) created with BioRender.com, released under a Creative Commons Attribution-NonCommercial-NoDerivs 4.0 International license.

by KEAP1[60], both have overlapping targets and roles in preventing liver-related diseases[61]. Further, NRF1 has been shown to be important for the survival of hepatocytes and to mediate oxidative stress during development[62,63]. Thus, we examined if the observed dispensable role of NRF2 in the absence of GSH was due to a compensatory activity by NRF1. We observed that livers from Gclc L-KO mice had increased expression of NRF1 protein (Fig. S12A). Interestingly, the levels of NRF1 in the livers of L-DKO mice were lower than in the Gclc L-KO mice; however, they remained higher than control mice. These data suggest that GSH synthesis, potentially through NRF2 repression, also limits NRF1 protein expression in the liver. Further, these data suggest that at acute time points following *Gclc* genetic deletion, compensation by NRF1 could be responsible for the dispensable role of NRF2 in GSH-depleted conditions. Further research is required to fully elucidate the interplay of NRF1 and NRF2 following GSH depletion in the liver.

Next, we evaluated the impact of individual and combined liver-specific Gclc/Nrf2 deletions on markers of steatohepatitis. Pro-fibrotic markers yes-associated protein 1 (YAP1) and taffazin (TAZ) and autophagy marker p62 showed elevated expression in Gclc L-KO mice (Fig. S12B, S12C). Interestingly, these markers largely reversed in liver-specific Gclc-Nrf2 DKO mice. These data suggest that while markers of fibrosis and autophagy are increased following liver-specific Gclc deletion, these are potentially due to the activation of NRF2. Additionally, no changes in markers of oxidative DNA damage (Fig. S13A) and lipid peroxidation (Fig. S13B) were observed in the L-DKO mice. These findings further suggest that at acute time points, both GCLC and NRF2 are dispensable to the liver.

Our results indicate that GSH supports lipid abundance by preventing NRF2 activation in the liver, as well as through other potential NRF2-independent mechanisms. NRF2 activation has been shown to repress the expression of lipogenic genes[64–67]. NRF2 protein stability is

regulated by KEAP1. Pharmacologically, KEAP1 activity can be inhibited by the compound bardoxolone (CDDO)[68]. Therefore, to further dissociate the effects of GSH and NRF2 on lipid abundance and assess if NRF2 activation alone mimics the effects of liver-specific Gclc deletion, we treated Gclc WT mice with CDDO-methyl (CDDO-Me), a form of bardoxolone with increased bioavailability. Treatment with CDDO-Me increased NRF2 expression but not to the level observed in the liver of Gclc L-KO mice, as we could only detect substantial NRF2 expression after enriching the whole protein lysate for a nuclear fraction (Fig. S14A, S14B). CDDO-Me treatment increased the expression of NRF2 target genes (Fig. S14C-S14D) and lowered the expression of lipogenic genes (Figs. 5K, S14C). Consequently, CDDO-Me treatment caused a reduction in serum triglycerides (Fig. 5L). However, these resulting effects of NRF2 induction via CDDO treatment were to a dramatically lower degree compared to the effects of NRF2 induction upon GSH depletion in Gclc L-KO mice (Figs. 4E, F & 5D, E). CDDO-Methyl treatment did not impact body weight, liver damage, or serum glucose and cholesterol levels (Fig. S14F–S14J). One hypothesis is that we observed a greater induction of NRF2 through liver-specific *Gclc* deletion as compared to KEAP1 inhibition with CDDO-me because CDDO-me is a weaker inhibitor of NRF2-KEAP1 interaction compared to the ROS generated following GSH depletion. Alternatively, GSH could have potentially prevented NRF2 activation through both KEAP1-independent mechanisms. Indeed, it was recently reported that treatment of cells with $H_2O_2$, which can lower GSH levels, leads to increased expression of NRF2 target genes in a KEAP1-independent mechanism[69]. Further research is required to fully elucidate the KEAP1-dependent and -independent mechanisms of GSH-dependent regulation of NRF2.

NRF2 is reported to have suppressive effects on the lipogenic pathway via inhibiting the activity of LXR[64,70], a transcription factor

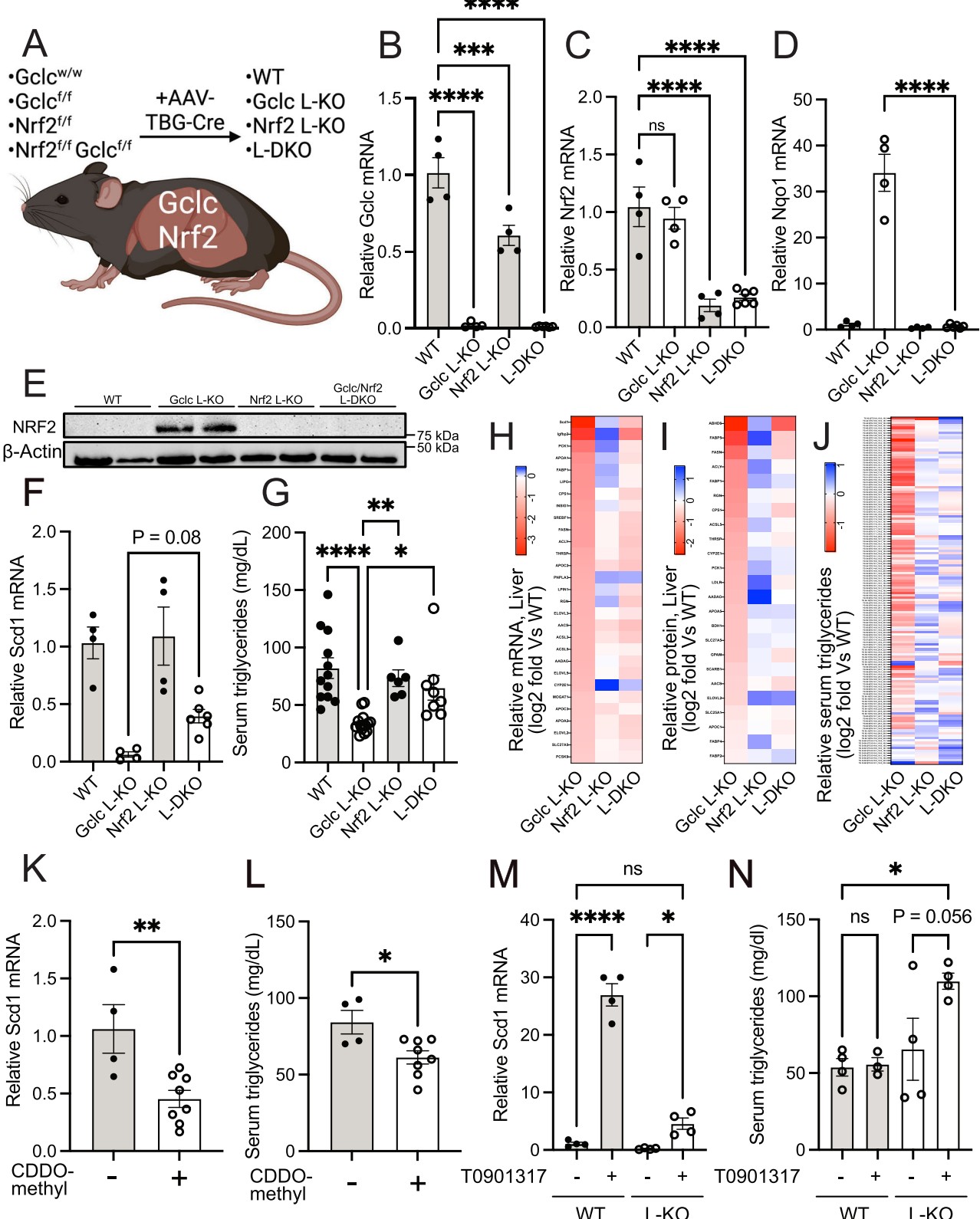

that supports lipid synthesis. To test if this mechanism was at play upon depleting GSH in the liver, we treated Gclc L-KO mice with T0901317, an LXR agonist. Induction of LXR activity in the Gclc L-KO mice rescued lipogenic gene expression and circulating triglycerides (Fig. 5M, N). These data suggest that NRF2 activation following GSH depletion causes repression of LXR activity, leading to lower expression of lipogenic enzymes and lower levels of serum triglycerides.

## Sustained depletion of NRF2 upon *Gclc* deletion induces a sex-dependent requirement for NRF2 to maintain liver homeostasis and animal survival

Liver-specific deletion of *Gclc* impaired the expression of lipogenic enzymes and lowered circulating triglyceride levels. However, fat stores (i.e., epididymal white adipose tissue) were not significantly depleted in liver-specific Gclc KO mice. Further, we found NRF2 to be dispensable to the liver upon GSH depletion. Notably, these

**Fig. 5 | GSH in the liver supports triglyceride levels in an NRF2-dependent manner. A** Schematic of inducible liver-specific *Gclc* deletion (Gclc L-KO), *Nrf2* deletion (Nrf2 L-KO), and *Gclc-Nrf2* deletion (L-DKO). Liver-specific knockout of genes was achieved using AAV-TBG-Cre. All mice were sacrificed 21 days post AAV-TBG-Cre treatment unless otherwise indicated. (B-D) Relative expression of (**B**) *Gclc*, (**C**) *Nrf2* and (**D**) *Nqo1* mRNA in the liver of WT ($n = 4$), Gclc L-KO ($n = 4$), Nrf2 L-KO ($n = 4$), and L-DKO ($n = 6$) mice following treatment with AAV-TBG-Cre. Expression levels were normalized to the expression of the reference gene *Rps9*. A one-way ANOVA with subsequent Dunnett's multiple comparisons test (used in (**B**) and (**C**)) or Šidák's multiple comparisons test (used in (**D**)) was used to determine statistical significance ((**B**): WT vs. Gclc L-KO P value < 0.0001, WT vs. Nrf2 L-KO *P* value = 0.0004, WT vs. L-DKO P value < 0.0001. **C** WT vs. Gclc L-KO *P* value = 0.7972, WT vs. Nrf2 L-KO *P* value < 0.0001, WT vs. L-DKO P value < 0.0001. **D** Gclc L-KO vs. L-DKO *P* value < 0.0001). **E** Representative immunoblot analysis of NRF2 in the liver of WT, Gclc L-KO, Nrf2 L-KO, and L-DKO mice following treatment with AAV-TBG-Cre. Data shown are representative of at least three replicates. **F** Relative expression of *Scd1* mRNA in the liver of WT ($n = 4$), Gclc L-KO ($n = 4$), Nrf2 L-KO ($n = 4$), and L-DKO ($n = 6$) mice following treatment with AAV-TBG-Cre. Expression levels were normalized to the expression of the reference gene *Rps9*. A one-way ANOVA with subsequent Šidák's multiple comparisons test was used to determine statistical significance (Gclc L-KO vs. L-DKO *P* value = 0.0866). **G** Serum triglyceride levels for WT ($n = 12$), Gclc L-KO ($n = 14$), Nrf2 L-KO ($n = 6$), and L-DKO ($n = 8$) mice 3 weeks post-treatment with AAV-TBG-Cre. A one-way ANOVA with subsequent Dunnett's multiple comparisons test was used to determine statistical significance (Gclc L-KO vs. WT P value < 0.0001, Gclc L KO vs. Nrf2 L-KO *P* value = 0.0040, Gclc L-KO vs. L-DKO *P* value = 0.01421). **H** Relative expression of lipogenic genes mRNA in the liver of WT ($n = 4$), Gclc L-KO ($n = 4$), Nrf2 L-KO ($n = 4$), and L-DKO ($n = 4$) mice

expressed as Log2 fold change. **I** Relative expression of proteins related to the lipogenic pathway in the liver of WT ($n = 4$), Gclc L-KO ($n = 4$), Nrf2 L-KO ($n = 4$) and L-DKO ($n = 4$) mice expressed as Log2 fold change. **J** Relative abundance of serum triglycerides in the liver of WT ($n = 4$), Gclc L-KO ($n = 4$), Nrf2 L-KO ($n = 4$) and L-DKO ($n = 4$) mice expressed as Log2 fold change. **K** Relative mRNA expression of *Scd1* in the liver of WT mice following a 4-day treatment with either vehicle ($n = 4$) or CDDO-methyl ($n = 8$). Expression levels were normalized to the expression of the reference gene *Rps9*. An unpaired two-tailed t-test was used to determine statistical significance (Vehicle vs. CDDO-methyl P value = 0.0065). **L** Triglyceride levels in the serum of WT mice without AAV-TBG-Cre injection following a 4-day treatment with either vehicle ($n = 4$) or CDDO-methyl ($n = 8$). An unpaired two-tailed *t* test was used to determine statistical significance (Vehicle vs. CDDO-methyl P value = 0.0171). **M** Relative expression of *Scd1* mRNA in the liver of WT ($n = 4$) and L-KO ($n = 4$) mice treated with either vehicle or T0901317 on days 17–20 post AAV-TBG-Cre injection. Expression levels were normalized to the expression of the reference gene *Rps9*. A one-way ANOVA with subsequent Šidák's multiple comparisons test was used to determine statistical significance (vehicle vs. T0901317: WT *P* value < 0.0001, L-KO *P* value = 0.0462; WT/Vehicle vs. L-KO/T0901317 *P* value = 0.1254). **N** Triglyceride levels in the serum of WT ($n = 4$) and L-KO ($n = 4$) mice treated with either vehicle or T0901317 on days 17–20 post AAV-TBG-Cre injection. A one-way ANOVA with subsequent Šidák's multiple comparisons test was used to determine statistical significance (vehicle vs. T0901317: WT P value = 0.9994, L-KO *P* value = 0.0560; WT/Vehicle vs. L-KO/T0901317 *P* value = 0.0155). Indicated *n* values represent biologically independent samples from mice. Data are shown as mean ± SEM. ns = not significant, *P value < 0.05, **P value < 0.01, ***P value < 0.001, ****P value < 0.0001. (A) created with BioRender.com, released under a Creative Commons Attribution-NonCommercial-NoDerivs 4.0 International license.

---

observations were found in a relatively short time after *Gclc* deletion (i.e., three weeks). To test the effect of sustained repression of lipogenic programs and a concurrent sustained induction of NRF2 activity, we monitored mice for ten weeks following the induction of *Gclc* deletion alone (Gclc L-KO) or in combination with *Nrf2* deletion (L-DKO) in the liver (Fig. 6A). Nearly all liver-specific Gclc L-KO mice survived over time (Fig. 6B). However, we found the L-DKO mice had a reduced survival rate which was sex-specific (Fig. 6B, C). Male mice underwent rapid loss of survival at extended time points, while female mice were largely unaffected. In addition, histological analysis revealed a significant difference in liver damage between female and male L-DKO mice (Fig. 6D, E). Indeed, we found male L-DKO mice to have a greater hepatocellular injury than female Gclc L-KO mice (Fig. 6D, E). Liver damage markers in the serum were also found to be significantly elevated in the serum of male L-DKO mice; however, they were not greater than levels from male Gclc L-KO mice (Fig. S15A, S15B). Importantly, these levels were still lower than those reported in the non-inducible liver-specific Gclc KO mouse strain[30]. Like the acute time points of liver-specific *Gclc* deletion, serum triglycerides were depleted in mice with prolonged liver-specific *Gclc* deletion and rescued with *Nrf2* deletion (Fig. 6F). This was found not to be sex-dependent, as the rescue was observed in both the female and male mice. (Fig. S15C). Unlike the acute setting, however, prolonged ablation of GSH synthesis in the liver resulted in decreased abundance of epididymal white adipose depots, a major storage tissue for triglycerides (Fig. 6G). This was predominant in the male Gclc L-KO mice and was not rescued with *Nrf2* deletion (Fig. S15D). Additionally, we found sex-dependent changes in serum levels of cholesterol and glucose (Fig. S15E, S15F). Further analysis revealed that male L-DKO livers had dramatically increased markers of apoptosis and oxidative DNA damage compared to female mice (Fig. S16). Interestingly, we found increased apoptotic markers in the adipose tissue of both female and male L-DKO mice compared to L-Gclc KO mice (Fig. S17); however, more research is required to fully elucidate the mechanisms involved in the adipose tissue of these mice. Together, these data reveal a sex-dependent requirement for NRF2 in the absence of GSH over time. Further, these data suggest that by maintaining circulating

triglycerides, GSH synthesis in the liver potentially supports fat stores in the body.

## Discussion

GSH has been linked to nearly every cellular process in the body[71,72]. However, its function in vivo remains poorly understood. To address this knowledge gap, we have developed a series of genetic models that permit spatiotemporal control over the expression of GCLC, the rate-limiting enzyme in GSH synthesis. We find that the loss of GCLC across all tissues of adult animals results in rapid weight loss and death. However, this does not cause immediate damage to multiple tissues, including the liver. Although we found higher levels of serum biomarkers of liver damage biomarkers in the Gclc L-KO mice compared to Gclc WT mice, these were not necessarily indicative of liver damage as they were dramatically lower than reported ALT/AST levels in mice with acute liver toxicity (>1500 U/L)[30,73,74]. This is unexpected since we found the liver to have higher levels of GSH than other tissues. Additionally, a previously studied liver-specific deletion of *Gclc* was shown to cause steatohepatitis and liver failure[30]. One difference between their model and the one reported here is that deletion of *Gclc* is induced in hepatocytes of adult mice (which are slowly proliferating), compared to theirs, where deletion of *Gclc* occurred in hepatocytes of growing animals (which are rapidly proliferating). The requirement of a protein pre- or post-organ and -animal development is an important distinction that is not always considered in studies involving tissue-specific knockout mouse models but potentially requires additional investigation.

A central question we ask here is, if GSH is not immediately required for liver survival, then why is it produced at such high levels? The clearest hypothesis is that GSH synthesis is required for liver function. Indeed, the liver has been reported to need a maximal synthesis of GSH for detoxifying exogenous xenobiotics, such as acetaminophen[75,76]. Further, GSH is described to buffer endogenous electrophiles[77,78]. We hypothesized that GSH synthesis plays additional detoxification-independent roles in liver function. The liver is known to be responsible for synthesizing new lipids. Also, lowering GSH levels in animals has previously been shown to prevent obesity-related phenotypes in mice, such as weight gain under a high-fat diet[79,80].

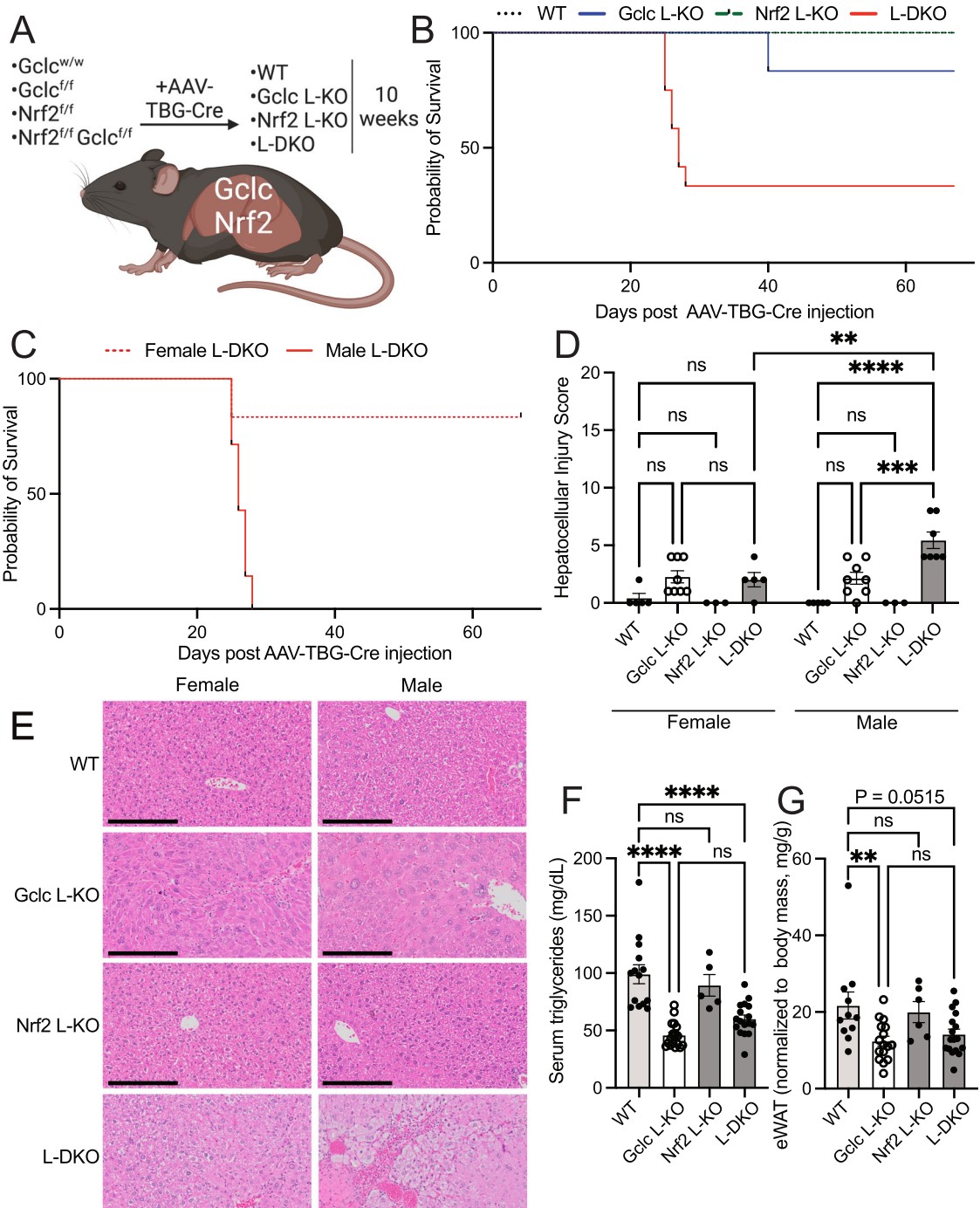

**Fig. 6 | Prolonged *Gclc* deletion in the liver induces a sex-dependent requirement of NRF2. A** Schematic for extended monitoring of *Gclc* deletion (Gclc L-KO), *Nrf2* deletion (Nrf2 L-KO), and *Gclc-Nrf2* deletion (L-DKO) in mice. Liver-specific gene knockout was induced with AAV-TBG-Cre. Mice were sacrificed 10 weeks post AAV-TBG-Cre injection unless humane endpoints were reached earlier. **B** Percent survival for WT (*n* = 8), Gclc L-KO (*n* = 6), Nrf2 L-KO (*n* = 6) and L-DKO (*n* = 12) mice following treatment with AAV-TBG-Cre. **C** Percent survival for female (*n* = 6) and male (*n* = 7) L-DKO mice following treatment with AAV-TBG-Cre. **D** Hepatocellular Injury Score from H&E-stained slides of the liver from female WT (*n* = 5), Gclc L-KO (*n* = 8), Nrf2 L-KO (*n* = 3), L-DKO (*n* = 4) and male WT (*n* = 5), Gclc L-KO (*n* = 8), Nrf2 L-KO (*n* = 3), L-DKO (*n* = 7) mice. A two-way ANOVA with subsequent Tukey's multiple comparisons test was used to determine statistical significance (Female: WT vs. Gclc L-KO P value = 0.2504, WT vs. Nrf2 L-KO P value = 0.9999, WT vs. L-DKO P value = 0.5546, Gclc L-KO vs. L-DKO P value > 0.9999; Male: WT vs. Gclc L-KO P value = 0.1238, WT vs. Nrf2 L-KO P value > 0.9999, WT vs. L-DKO P value < 0.0001, Gclc L-KO vs. L-DKO P value = 0.0006; Female L-DKO vs. Male L-DKO P value = 0.0020). **E** Representative H&E-stained slides of the liver from female and male

WT, Gclc L-KO, Nrf2 L-KO, and L-DKO mice 10 weeks following treatment with AAV-TBG-Cre. Scale bars = 200 μm. Data shown are representative of at least 3 replicates. **F** Serum triglyceride concentration of WT (*n* = 14), Gclc L-KO (*n* = 19), Nrf2 L-KO (*n* = 5) and L-DKO (*n* = 17) mice 10 weeks following treatment with AAV-TBG-Cre. A one-way ANOVA with subsequent Tukey's multiple comparisons test was used to determine statistical significance (WT vs. Gclc L-KO P value < 0.0001, WT vs. Nrf2 L-KO P value = 0.7838, WT vs. L-DKO P value < 0.0001, Gclc L-KO vs. L-DKO P value = 0.1525). **G** Epididymal fat adipose tissue (eWAT) mass normalized to body mass from WT (*n* = 10), Gclc L-KO (*n* = 16), Nrf2 L-KO (*n* = 6) and L-DKO (*n* = 17) mice 10 weeks post-treatment with AAV-TBG-Cre. A one-way ANOVA with subsequent Tukey's multiple comparisons test was used to determine statistical significance (WT vs. Gclc L-KO P value = 0.0112, WT vs. Nrf2 L-KO P value = 0.9675, WT vs. L-DKO P value = 0.0515, Gclc L-KO vs. L-DKO P value = 0.8936). Indicated *n* values represent biologically independent samples from mice. Data are shown as mean ± SEM. ns = not significant, * P value < 0.05, ** P value < 0.01, *** P value < 0.001, **** P value < 0.0001. (A) created with BioRender.com, released under a Creative Commons Attribution-NonCommercial-NoDerivs 4.0 International license.

In keeping with this observation, here we unveil a crucial role of GSH in lipid production, as the synthesis of GSH, specifically in the liver, is required for sustaining physiological circulating triglyceride levels (an important energy source) and adipose tissue homeostasis under standard dietary regimen. This is a significant phenotype, as a similar reduction in circulating triglycerides was not observed with a liver-specific loss of critical lipogenic enzymes, such as fatty acid synthase (FASN)[81]. Together, our findings indicate that GSH synthesis in the liver is a crucial contributor to lipid abundance in vivo.

Mechanistically, we show that, upon GSH depletion, the liver tissue induces the expression of target genes associated with the antioxidant transcription factor NRF2. Since previous studies have linked the activation of NRF2 with the repression of lipogenic gene expression[64], we reasoned that GSH depletion causes oxidation and consequent inactivation of KEAP1 (a repressor of NRF2), thus leading to increased antioxidant and decreased lipogenic gene expression. However, KEAP1 regulates multiple pathways beyond NRF2[60,82], and the contributions of NRF2 to lipid homeostasis have been reported to be pleiotropic and context-dependent[83]. Chronic GSH deficiency by *Gclm* deletion in mice has also been associated with metabolic alterations and the activation of the stress-responsive factor AMPK[53,84]. Further, AMPK has been reported to be an upstream activator of NRF2[85]. Nonetheless, we show that following GSH depletion, the repression of lipogenic gene expression and serum triglyceride levels is dependent on NRF2 activation. Further investigations are required to elucidate why and how NRF2 activation blocks lipid production. One possible scenario is the metabolic cofactor NADPH, which is known to be required not only for lipid synthesis but also for the function of antioxidant reductases, such as glutathione reductase (GSR) and thioredoxin reductase (TXNRD)[86]. Notably, the deletion of lipogenic enzymes in the liver, such as acetyl-CoA carboxylase 1/2 (ACC1/2), has been reported to liberate NADPH for antioxidant processes, including GSH regeneration[87].

In many scenarios, activation of NRF2 has been reported to be a stress response that is required to prevent the accumulation of oxidative stress and subsequent tissue damage. Surprisingly, we demonstrate here that, in an acute setting, mice with combined liver-specific deletion of *Gclc* and *Nrf2* do not lose weight or show poor health or liver failure. This suggests that the antioxidant response provided by NRF2 is not required for liver survival in the absence of GSH. Instead, our findings demonstrate an important role for GSH in the liver, where under resting conditions, GSH maintains low NRF2 activity to permit ample expression of lipogenic enzymes, promotes the production of triglycerides, and sustains lipid abundance in vivo.

Although prolonged GSH depletion induced triglyceride depletion and reduced epididymal adipose tissue in mice, it induced a sex-dependent requirement of NRF2 expression in the liver. Indeed, several studies have reported some level of sex-dependent differences in susceptibility to liver damage. In mouse and rat models of choline and methionine-deficient diet-induced metabolic dysfunction-associated steatotic liver disease (MASLD), females are reported to be more protected than their male counterparts[88]. These observed sex-dependent differences have been attributed to sex-dependent metabolic alterations such as the estrogen-dependent synthesis of phosphatidylcholines[88–91]. Further, in fructose HFD-induced models, ovariectomy has been shown to reverse the reduced susceptibility of female mice to liver damage[92]. Further research is required to determine the underlying cause of sex-dependent liver damage upon extended loss of GCLC and NRF2. However, these basic findings potentially provide clues for sex-dependent etiologies of diseases that involve oxidative stress.

## Methods

### Experimental model and details

**Animal studies.** All animal studies were performed according to protocols approved by the University Committee on Animal Resources at the University of Rochester Medical Center. *Gclc*<sup>f/f</sup> mice were generated as described[30] and crossed with the Rosa26-CreERT2 mouse strain (Jackson Labs, #008463) or *Nrf2*<sup>f/f</sup> mouse strain (Jackson Labs, #025433). Animals were housed in rooms with a relative humidity range of 30-70% and temperature of 64–79 °F under a 12 h light/12 h dark cycle with ad libitum access to food and water. All animals were aged for at least 12 weeks before being used in their respective experiments. Tamoxifen (Sigma Aldrich, T5648) was administered by intraperitoneal injection at 160 mg/kg once daily for 5 days. For liver-specific genetic deletions, mice were injected via the tail vein with $2.5 \times 10^{11}$ GC of AAV-TBG-Cre (Addgene, 107787-AAV8) in PBS. For high-fat diet experiments, mice were fed a 60 kcal% fat high-fat diet (Research Diets, D12451i) following either tamoxifen or AAV-TBG-Cre injection for 2-3 weeks. Control mice were fed normal rodent chow (LabDiet, 5053). For CDDO-methyl experiments, WT mice without AAV-TBG-Cre injections were injected intraperitoneally with 5 mg/kg of either vehicle (10%DMSO in 90% corn oil) or CDDO-methyl (Med-Chem Express, HY-13324) for 4 days. For T0901317 experiments, mice were injected intraperitoneally with either vehicle (10%DMSO in 90% corn oil) or T0901317 (MedChem Express, HY-10626) at a dose of either 16.9 mg/kg or 50 mg/kg dose for either 4 days (17–20) days post AAV-TBG-Cre treatment) or 14 days (7–20) days post AAV-TBG-Cre treatment). With the GSH-ee experiments, mice were injected intraperitoneally with either vehicle (saline) or Glutathione ethyl ester (Cayman Chemical, 14953) for 4 days ((17-20) days post AAV-TBG-Cre treatment).

**Immunoblot assays.** Tissue samples were crushed on dry ice to obtain homogenous aliquots and then lysed in RIPA buffer (Thermo Scientific #89900) containing Halt protease & phosphatase inhibitor (Thermo Scientific #1861280). Extracted proteins were quantified using the Pierce BCA Protein Assay Kit (Thermo Scientific #23225). 75 μg of protein lysates were heated for 10 min at 95 °C in Laemmli 6X SDS sample buffer (Boston BioProducts, # BP-111R) with 5% 2-mercaptoethanol (VWR Life Science #M131-100ml) and then ran on 4–20% Criterion TGX pre-cast gels (Bio-Rad #5671093). Separated proteins were transferred onto Immobilon-P Transfer membranes (MilliporeSigma #IPVH00010), blocked for an hour using 5% milk in TBST, and stained overnight with targeting primary antibodies in 5% milk TBST. Stained membranes were then washed with TBST and stained with corresponding secondary antibodies in 5% milk in TBST for one hour. Antibody-stained protein signal was amplified and visualized using SuperSignal West Pico PLUS chemiluminescent substrate (Thermo Fisher #34578) and imaged with a ChemiDoc MP Imaging system (Bio-Rad). The antibodies used for the immunoblot assays were: GCLC (Santa Cruz Biotech, #sc-390811), NQO1 (Sigma Prestige Antibodies, HPA007308), ACTIN (Sigma, A1978), NRF2 (Cell Signaling Technology, #12721), NRF1 (Cell Signaling Technology, #8052), YAP/TAZ (Cell Signaling, #8418) 4-HNE (Abcam, #ab46545) and Lamin A/C (Cell Signaling, #4777).

**RNA analysis.** Tissue samples were crushed on dry ice and homogenized using a bead Mill (VWR). mRNA was then isolated from the tissues using E.Z.N.A. total RNA Kit I (Omega Bio-Tek, R6834-02). For gene expression analysis, 1 μg of RNA was used for cDNA synthesis using qScript cDNA Synthesis Kit (Quanta Bio, #66196756). The expression of target genes was analyzed via quantitative real-time (RT) PCR with a QuantaStudio 5 qPCR machine (Applied Biosystems, Thermo Fisher Scientific). Primers used for quantitative RT PCR analysis are included in Supplementary Data S6.

**RNA-seq analysis.** RNA-seq was performed on tissues using Genomics Research Center (GRC) at URMC and the Bauer Core Facility at Harvard University. Differential expression and GSEA analyses were performed as previously described[93,94].

**Serum analysis.** Mice were anesthetized with isoflurane, after which blood was collected via the retro-orbital venous sinus into BD microtainer tubes (BD #365967). Serum was isolated from the blood by centrifuging blood samples at $10,000 \times g$ for 5 min. Analysis of liver damage biomarkers and serum triglycerides was carried out by VRL Animal Health Diagnostics.

**Tissue staining.** Five-micron formalin-fixed, paraffin-embedded tissue sections were used for hematoxylin and eosin (H&E) staining immunohistochemistry and immunofluorescence analyses. The tissues were dewaxed and rehydrated through a series of xylene and ethanol changes. For antibody staining, antigen retrieval was performed on the slides by incubating them in a steamer for 40 min in citrate buffer (Vector Labs, Cat# H-3300-250). The slides were washed in water/PBS and then blocked using 5% goat serum in PBS for 1 h at room temperature prior to adding primary antibodies diluted in the blocking buffer. Primary antibody (NQO1; Abcam Cat# ab196196, NRF2; Abcam Cat# ab31163, TUNEL; Abcam; ab206386, Cleaved Caspase 3; Cell Signaling Cat# 9664, F4/80; Cell Signaling Cat# 70076, 8-oxoguanine; Abcam Cat# Ab62623, p62; Cell Signaling Cat# 23214) incubation was carried out overnight at 4 °C. For immunohistochemistry, detection was performed using a biotinylated anti-rabbit IgG secondary antibody and streptavidin-Horseradish peroxidase (HRP), followed by colorimetric detection using DAB. The sections were then counterstained with hematoxylin. For immunofluorescence, the second-to-last wash included 1 µg/ml of 4′, 6′-diamidino-2-phenylindole dihydrochloride (DAPI; Sigma, Cat# D9542-10mg) in PBS to stain cell nuclei. Immunofluorescence tissues were mounted with ProLong Gold (Life Technologies, Cat# P36934), while H&E and immunohistochemistry tissues were mounted with Permount mounting media (Fisher Scientific, Cat# SP15500) and coverslipped for imaging. Immunofluorescence-stained slides were imaged using a CyteFinder II from Rarecyte. H&E and immunohistochemistry slides were imaged using an Olympus VS120 virtual slide microscope and Visiopharm image analysis system.

**Preparation of NEM-derivatized cysteine and GSH internal standards.** The N-ethylmaleimide (NEM) derivatized isotope labeled [$^{13}C_3$, $^{15}N$]-cysteine-NEM and [$^{13}C_2$,$^{15}N$]-GSH-NEM were prepared by derivatizing the [$^{13}C_3$, 15N]-cysteine and [$^{13}C_2$,$^{15}N$]-GSH standards with 50 mM NEM in 10 mM ammonium formate (pH = 7.0) at room temperature (30 min) as previously described[95]. [$^{13}C_4$, $^{15}N_2$]-GSSG was prepared from the oxidation of [$^{13}C_2$,$^{15}N$]-GSH as described[96].

**Non-targeted lipidomics—sample preparation.** The liver, brain, kidney, lung, and adipose tissue were homogenized with a pre-chilled BioPulverizer (59012MS, BioSpec) and then placed on dry ice. The chloroform:methanol extraction solvent (v:v = 1:1) was added to the homogenate to meet 50 mg/mL. The samples were then sonicated in ice-cold water using Bioruptor UCD-200 sonicator for 5 min (30 s sonication and 30 s rest cycle; high voltage mode). The lipid extracts were cleared by centrifugation ($17,000\ g$, 20 °C, 10 min), and the lipids in the supernatant were analyzed by LC-MS.

For serum samples, 75 µL of chloroform:methanol extraction solvent (v:v = 1:2) was added to 20 µL of mouse serum, with the exception of the serum from WT, Gclc L-KO, Nrf2 L-KO, and L-DKO mouse serum, for which 20 µL of mouse serum was combined with 180 µL of chloroform:methanol extraction solvent (v:v = 1:2) containing internal standards at the final concentrations: 5 nM D7-Sphinganine (Avanti Polar Lipids Inc., Cat# 860658), 12.5 nM D3-Deoxysphinganine (Avanti Polar Lipids Inc., Cat# 860474), and SPLASH LIPIDOMIX (1:1000, Avanti Polar Lipids Inc., Cat# 330707). After sonicating (1400 rpm, 20 °C, 5 min), the extracts were cleared by centrifugation ($17,000\ g$, 20 °C, 10 min), and the lipids in the supernatant were analyzed by LC-MS.

**Non-targeted lipidomics—instrumental condition and data analysis.** The HPLC conditions were identical to the previous study[97]. In brief, lipidomics for each sample were performed by the Vanquish UPLC systems coupled to a Q Exactive HF (QE-HF) mass spectrometer equipped with HESI (Thermo Fisher Scientific, Waltham, MA). Chromatographic separation was conducted on a Brownlee SPP C18 column (2.1 mm × 75 mm, 2.7 µm particle size, Perkin Elmer, Waltham, MA) using mobile phase A (100% H2O containing 0.1% formic acid and 1% of 1 M NH4OAc) and B (1:1 acetonitrile:isopropanol containing 0.1% formic acid and 1% of 1 M NH4OAc). The gradient was programmed as follows: 0–2 min 35% B, 2–8 min from 35 to 80% B, 8–22 min from 80 to 99% B, 22–36 min 99% B, and 36.1–40 min from 99 to 35% B. The flow rate was 0.400 mL/min.

For the mass spectrometry, the data-dependent MS$^2$ scan conditions were applied in both positive and negative mode: the scan range was from m/z 250–1500, resolution was 60,000 for MS, and 30,000 for DDMS$^2$ (top 10), and the AGC target was 3E$^6$ for full MS and 1E$^5$ for DDMS$^2$, allowing ions to accumulate for up to 200 ms for MS and 50 ms for MS/MS. For MS/MS, the following settings are used: isolation window width 1.2 m/z with an offset of 0.5 m/z, stepped NCE at 10, 15, and 25 a.u., minimum AGC 5E$^2$, and dynamic exclusion of previously sampled peaks for 8 s.

For the analysis of lipids in the serum from WT, Gclc L-KO, Nrf2 L-KO, and L-DKO mice, the methods were the same with the following exceptions: MS2 scan conditions were applied in positive mode, the scan range was from m/z 120-1000, and the resolution was 120,000 for MS. For the MS/MS scan the following conditions were used: NCE at 20, 30, and 40 a.u. Quality control (QC) samples were included to check the technical variability and were prepared by mixing an equal volume of lipid extract from each tissue or serum sample. QC samples were included in the analysis sequence every ten samples and monitored for changes in peak area, width, and retention time to determine the performance of the LC-MS/MS analysis. QC samples were subsequently used to align the analytical batches.

The lipid peaks were identified, aligned, and exported using MS-DIAL[98]. The data were further normalized to the median value of total lipid signals. Only lipids fully identified by MS$^2$ spectra were included in the analysis.

**Mass spectrometry analysis of metabolites—sample preparation.** The liver, brain, kidney, and lung tissues were homogenized with a pre-chilled BioPulverizer (59012MS, BioSpec) and then placed on dry ice. For the non-targeted metabolite analysis, the tissue metabolites were extracted in 80% MeOH at a final tissue concentration of 50 mg/mL for 24 h at −80 °C. For the quantification of sulfur metabolites in the tissue samples, the tissue metabolites were extracted and derivatized with NEM in ice-cold extraction solvent (80% MeOH: 20% H2O containing 25 mM NEM and 10 mM ammonium formate, pH=7.0) which includes stable isotope labeled internal standards (20 µM [$^{13}C_3$, $^{15}N$]-cysteine-NEM, 36.4 µM [$^{13}C_2$, $^{15}N$]-GSH-NEM, 10 µM [D$_4$]-Cystine, 0.92 µM [$^{13}C_5$, $^{15}N_2$]-GSSG, 20 µM [D$_4$]-Hypotaurine, and 20 µM of [$^{13}C_2$]-Taurine at a final concentration of 50 mg/mL followed by incubation on 4 °C for 24 hr.

For the global metabolite profiling in serum, the metabolites in 10 µL of serum were extracted by the 390 µL of pre-chilled 82% MeOH (−80 °C) followed by 15 min incubation at −80 °C. For the quantification of sulfur metabolites in serum, the metabolites in 20 uL of serum were extracted and derivatized by NEM in 80 µL of ice-cold extraction solvent (80% MeOH: 20% H2O containing 25 mM NEM and 10 mM ammonium formate, pH = 7.0) which includes stable isotope labeled internal standards (1 µM [$^{13}C_3$, $^{15}N$]-cysteine-NEM, 0.1 µM [$^{13}C_2$, $^{15}N$]-GSH-NEM, 2 µM [D$_4$]-Cystine, 4.6 µM [$^{13}C_5$, $^{15}N_2$]-GSSG, 5 µM [D$_4$]-Hypotaurine, and 40 µM [$^{13}C_2$]-Taurine followed by incubation at 4 °C for 30 min.

After centrifugation (17,000 $g$, 20 min, 4 °C), all the supernatants were analyzed by LC-HRMS.

**Mass spectrometry analysis of metabolites—instrumental condition and data analysis.** For the mass spectrometry analysis of metabolites, the previously established LC-MS conditions were applied[99]. For the chromatographic metabolite separation, the Vanquish UPLC systems were coupled to a Q Exactive HF (QE-HF) mass spectrometer equipped with HESI (Thermo Fisher Scientific, Waltham, MA). Samples were run on either a SeQuant ZIC-pHILIC LC column, 5 μm, 150 × 4.6 mm (MilliporeSigma, Burlington, MA) with a SeQuant ZIC-pHILIC guard column, 20 ×4.6 mm (MilliporeSigma, Burlington, MA) or an Atlantis Premier BEH Z-HILIC VanGuard FIT column, 2.5 μm, 2.1 mm × 150 mm (Waters, Milford, MA). For all samples, mobile phase A was 10 mM $(NH_4)_2CO_3$ and 0.05% $NH_4OH$ in $H_2O$, while mobile phase B was 100% ACN. The column chamber temperature was set to 30 °C. The mobile phase condition was set according to the gradient of 0-13 min: 80% to 20% of mobile phase B, 13–15 min: 20% of mobile phase B. The ESI ionization mode was positive and negative. For samples run on the ZIC-pHILIC column, the MS scan range (m/z) was set to 60–900. For those run on the BEH X-HILIC column, the MS scan range was set to 65–975. The mass resolution was 120,000, and the AGC target was 3 ×10⁶. The capillary voltage and capillary temperature were set to 3.5 KV and 320 °C, respectively. 5 μL of the sample was loaded. The LC-MS metabolite peaks were manually identified and integrated by EL-Maven (Version 0.11.0) by matching with a previously established in-house library[99].

For the targeted sulfur metabolite quantification approach, the previously established LC-MS conditions were applied[99] with selected reaction monitoring (MRM) using an Ultimate 3000 UPLC system coupled to a Thermo Finnigan TSQ Quantum equipped with HESI (Thermo Fisher Scientific, Waltham, MA). As a stationary phase, an XBridge Amide Column 3.5 μm (2.1 × 100 mm) (Waters, Milford, MA) was used. The mobile phase A was 97% water and 3% ACN (20 mM $NH_4Ac$, 15 mM $NH_4OH$, pH = 9.0), and the mobile phase B was 100% ACN. The column temperature was set to 40 °C, and the gradient elution was at 0.35 mL/mL of flow rate: 0 to 3 min, linear gradient from 15% to 70% of Phase A; 3–12 min: linear gradient from 70% to 98% of Phase A; 12 to 15 min, sustaining 98% of Phase A. The MS acquisition operated in the positive or negative mode. The capillary temperature was 305 °C, and the vaporizer temperature was 200 °C. The sheath gas flow was 75, and the auxiliary gas flow was 10. The spray voltage was 3.7 kV. The MRM conditions (parent ion → fragment ion; collision energy) of metabolites were as follows. Positive mode: Cysteine-NEM (m/z 247 → m/z 158; 30); [$^{13}C_3$, $^{15}N$]-Cysteine-NEM (m/z 251 → m/z 158; 30); GSH−NEM m/z (m/z 433 → m/z 304; 15); [$^{13}C_2$,$^{15}N$]-GSH-NEM (m/z 436 → 307 m/z; 15); Cystine (m/z 241→ m/z 74; 30); [$D_4$]-Cystine (m/z 245 → m/z 76; 30); GSSG (m/z 613 → m/z 355; 25), [$^{13}C_4$, $^{15}N_2$]-GSSG (m/z 619 → m/z 361; 25); Hypotaurine (m/z 110 → m/z 92; 10); [$D_4$]-Hypotaurine (m/z 114 → m/z 96; 10); Taurine (m/z 126 → m/z 108; 11); [$^{13}C_2$]-Taurine − (m/z 128 → m/z 110; 11). All peaks were manually integrated using Thermo Xcaliber Qual Browser. The quantification of metabolites was calculated by an isotope ratio-based approach according to published methods[100].

**Global protein abundance profiling by mass spectrometry**
**Sample preparation (A).** Flash frozen liver sections (~25 mg) from WT or *Gclc*⁻/⁻ mice (3 biological replicates each per TMT sample) were thawed on ice and resuspended in DPBS, supplemented with protease (cOmplete, EDTA-free protease inhibitor cocktail, Roche, #11873580001) and phosphatase (PhosSTOP, Roche, #4906845001) inhibitor tablets. Tissue was homogenized by probe sonication (2 × 10 pulses, 40% power output), and particulate matter was removed by passing samples through 0.4 μm syringe filters. The proteome concentration of the tissue lysates was determined using the DC protein assay (Bio-Rad), normalized to 2 mg/mL, and then 100 μL of each sample was transferred to a LoBind Eppendorf tube containing 48 mg of urea. Samples were reduced with DTT (5 μL of 200 mM stock in $H_2O$, 10 mM final concentration) and incubated at 65 °C for 15 min, then alkylated with iodoacetamide (5 μL of 400 mM stock in $H_2O$, 20 mM final concentration) and shaken at 37 °C for 30 min in the dark. Ice-cold MeOH (600 μL), $CHCl_3$ (200 μL), and $H_2O$ (500 μL) samples were vortexed and then centrifuged (10,000 $g$, 10 min, 4 °C) to precipitate proteins. The upper layer of supernatant was removed, and ice-cold MeOH (600 μL) was added to wash the protein disc. Samples were vortexed again, centrifuged (16,000 $g$, 10 min, 4 °C) and then all supernatant was removed to leave a protein pellet. Samples were resuspended in 160 μL EPPS buffer (200 mM, pH 8.0) using a probe sonicator (1 × 10−15 pulses, power output 20%) and then digested with LysC (4 μL of 0.5 μg/μL per sample, resuspended in HPLC grade water, Wako-chemicals, Fujifilm #125-05061) for 2 h at 37 °C in a shaker incubator. Samples were then digested with trypsin (11 μL of 0.5 μg/μL per sample, resuspended in trypsin resuspension buffer containing 20 mM $CaCl_2$; Promega, #V542A) overnight at 37 °C in a shaker incubator. After incubation with trypsin, the peptide concentration in samples was estimated using a Micro BCA Protein Assay (Thermo Scientific, #23235), and a volume corresponding to 25 μg peptides was transferred to a new low-bind Eppendorf tube per sample. Sample volumes were normalized to 35 μL with EPPS buffer (200 mM, pH 8), diluted with HPLC grade $CH_3CN$ (9 μL), and then labeled (5 μL of 20 μg/μL per sample) with the corresponding TMTsixplex Isobaric Mass Tag (Themo Scientific, #90064B). Samples were incubated at room temperature for 1 h, vortexing intermittently, and then quenched by the addition of hydroxylamine (5 μL of 5% w/v in HPLC water per sample) and incubated for 15 min at room temperature. Samples were then acidified with formic acid (2.5 μL), and 2 μL of each sample was combined in a LoBind Eppendorf and dried using a Speedvac to perform a ratio check. Remaining samples were stored at −80 °C until after experimentally determining TMT channel intensities.

**Sample preparation (B).** In an alternate protocol, the frozen liver tissue was directly lysed by probe sonication (2 × 10 pulse, 40% power output) in 4 M urea/DPBS and briefly cleared of debris by centrifugation (5000 $g$, 5 min, 4 °C). The proteome concentration of lysates was normalized to 2 mg/mL, and 100 μL was transferred to a LoBind Eppendorf tube containing 48 mg of urea. Samples were reduced with DTT and alkylated with iodoacetamide as described above, then diluted with DPBS (300 μL), and taken forward to LysC and trypsin digestion without precipitation. Digested samples were desalted using a C18 spin column (Pierce C18 Spin, #89873) according to manufacturer instructions prior to resuspension in EPPS buffer and $CH_3CN$ for TMT labeling as described above.

**TMT ratio check.** The combined and dried "ratio check" sample was redissolved in Buffer A (5% $CH_3CN$, 95% water, 0.1% formic acid, 20 μL) and desalted using C18 stage tips prepared in-house using 3× C18 discs (3 M Empore) stacked in 200 μL pipette tips. Stage-tips were activated with MeOH (2 × 60 μL), washed with Buffer B (80% $CH_3CN$, 20% water, 0.1% formic acid) (1 × 60 μL), and equilibrated with Buffer A (2 × 60 μL). The entire sample was then passed through the stagetip twice before being eluted into a new LoBind Eppendorf using 80 μL of 70% $CH_3CN$/30% $H_2O$/0.1% formic acid. The desalted sample was evaporated to dryness in a speedvac and then resuspended in 10 μL Buffer A and analyzed by mass-spectrometry using the following LC-MS gradient: 5% buffer B in buffer A from 0-15 min, 5–15% buffer B from 15–17.5 min, 15–35% buffer B from 17.5–92.5 min, 35–95% buffer B from 92.5–95 min, 95% buffer B from 95–105 min, 95–5% buffer B from 105–107 min, and 5% buffer B from 107–125 min; and standard MS3-based quantification described below. Ratios were determined from the average peak intensities corresponding to each channel. After experimentally

determining TMT channel signal intensities, frozen samples were thawed, and a volume corresponding to 12.5 µg/sample was combined in a new LoBind Eppendorf tube and dried using a SpeedVac.

**High pH fractionation-.** TMT labeled, combined, and dried samples were resuspended in 300 µL of Buffer A (5% v/v MeCN, 95% v/v $H_2O$, 0.1% v/v formic acid) and fractionated by centrifugation using a peptide desalting column (Pierce, #89852). Briefly, the storage solution was removed (5000 $g$, 2 min), and columns were washed with $CH_3CN$ (2 ×300 µL, 5,000 $g$, 2 min), then equilibrated with buffer A (2 × 300 µL, 5000 $g$, 2 min). Resuspended samples were loaded onto equilibrated spin columns (2000 $g$, 2 min), passing the entire sample twice through the column. The column was then washed with buffer A (300 µL, 2,000 $g$, 2 min), and 10 mM aqueous $NH_4HCO_3$ containing 5% $CH_3CN$ (300 µL, 2000 $g$, 2 min), before peptides were eluted from the column as 15 fractions using 300 µL buffer containing an increasing concentration of $CH_3CN$ in 10 mM $NH_4HCO_3$ (%$CH_3CN$ = 7.5, 10, 12.5, 15, 17.5, 20, 22.5, 25, 27.5, 30, 35, 40, 45, 50, 75) (2000 $g$, 2 min). Every 5th fraction was combined into a new Eppendorf to make five final fractions that were dried using a SpeedVac vacuum concentrator. The resulting fractions were then re-suspended in buffer A (16 µL) and analyzed on an Orbitrap Fusion mass spectrometer.

**Data processing–.** Protein abundance was calculated as a ratio of *Gclc*$^{-/-}$ vs. WT samples for each peptide-spectra match by dividing each TMT reporter ion intensity by the average intensity for the channels corresponding to WT control (3 per TMT sample). Peptide-spectra matches were then grouped based on protein ID, excluding peptides with summed reporter ion intensities for the WT channels <15,000, coefficient of variation for WT channels >0.5, non-unique or non-tryptic peptide sequences. TMT reporter ion intensities were normalized to the median summed signal intensity across channels, and the data were filtered to retain proteins with at least two distinct peptides. The fold change in protein abundance in KO vs. WT mice was calculated for individual replicates, averaged, and converted to a log2 scale. Change in proteins abundance ≥1 log unit and $p$ value < 0.05 were considered significant.

**TMT liquid chromatography-mass-spectrometry (LC-MS) analysis**
Samples were analyzed by liquid chromatography-tandem mass spectrometry using an Orbitrap Fusion Tribrid Mass Spectrometer (Thermo Scientific) coupled to an UltiMate 3000 Series Rapid Separation LC system and autosampler (Thermo Scientific Dionex). The peptides were eluted onto an EASY-Spray HPLC column (Thermo ES902, ES903) using an Acclaim PepMap 100 (Thermo 164535) loading column and separated at a flow rate of 0.25 µL/min. Data were acquired using an MS3-based TMT method using the following scan parameters: scan sequence began with an MS1 master scan (Orbitrap analysis, resolution 120,000, 400−1700 m/z, RF lens 60%, automatic gain control [AGC] target 2E5, maximum injection time 50 ms, centroid mode) with dynamic exclusion enabled (repeat count 1, duration 15 s). The top ten precursors were then selected for MS2/MS3 analysis. MS2 analysis consisted of quadrupole isolation (isolation window 0.7) of precursor ion followed by collision-induced dissociation (CID) in the ion trap (AGC 1.8E4, normalized collision energy 35%, maximum injection time 120 ms). Following the acquisition of each MS2 spectrum, synchronous precursor selection (SPS) enabled the selection of up to 10 MS2 fragment ions for MS3 analysis. MS3 precursors were fragmented by HCD and analyzed using the Orbitrap (collision energy 55%, AGC 1.5E5, maximum injection time 120 ms, resolution was 50,000). For MS3 analysis, we used charge state−dependent isolation windows. For charge state $z$ = 2, the MS isolation window was set at 1.2; for $z$ = 3−6, the MS isolation window was set 0.7. The RAW files were uploaded to Integrated Proteomics Pipeline (IP2) and searched using the ProLuCID

algorithm (publicly available at http://fields.scripps.edu/yates/wp/?page_id=821) using a reverse concatenated, non-redundant version of the Mouse UniProt database (release 2017). Cysteine residues were searched with a static modification for carboxyamidomethylation (+57.02146 Da), and N-termini and lysine residues were searched with a static modification corresponding to the TMT tag (+229.1629 Da). Methionine residues were searched with a differential modification for oxidation (+15.9949 Da) and a maximum of four differential modifications were allowed per peptide. Peptides were required to be at least 5 amino acids long and fully tryptic. ProLuCID data was filtered through DTASelect (version 2.0) to achieve a peptide false-positive rate below 1%. The MS3-based peptide quantification was performed with reporter ion mass tolerance set to 30 ppm with Integrated Proteomics Pipeline (IP2).

### Statistical analysis
All statistical analysis was completed using either R or GraphPad Prism 9/10.

### Reporting summary
Further information on research design is available in the Nature Portfolio Reporting Summary linked to this article.

## Data availability
Data supporting these findings are included within the article and its supplementary material or from the corresponding author upon request. The mass spectrometry proteomics data have been deposited to the ProteomeXchange Consortium via the PRIDE[101] partner repository with the dataset identifier PXD052674. The RNA-seq data generated in this study have been deposited in the Gene Expression Omnibus (GEO) database under accession codes GSE263190 and GSE263119. The lipidomics data generated in this study have been deposited in Metabolomics Workbench under Project ID PR002013[102]. Source data are provided in this paper.

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

## Acknowledgements

We thank Jonathan Coloff and Samuel McBrayer for their feedback and discussions. We would also like to thank the Genomics Research Center (GRC), the Histology, Biochemistry, and Molecular Imaging (HBMI) Core at the Center for Musculoskeletal Research (CMSR), and the Center for Advanced Research Technologies (CART) at URMC, the Bauer Core Facility at Harvard University, and Proteomics/Metabolomics Core at Moffitt Cancer Center, which is funded in part by Moffitt's Cancer Center Support Grant (P30CA076292). We also thank Joan Brugge and the Ludwig Cancer Center at Harvard Medical School for their support. This work was supported by the Wilmot Cancer Institute Predoctoral Fellowship (G.A.), Wilmot Cancer Institute Pilot Funding (I.S.H.), the American Association for Cancer Research and Breast Cancer Research Foundation (20-20-26-HARR) (I.S.H.), Breast Cancer Coalition of Rochester (I.S.H.), a Sir Henry Wellcome Postdoctoral Fellowship (M.E.K.), and a FNR-CORE grant C21/BM/15796788 (D.B.), and NIH grants R01CA269813 (I.S.H.), R37CA230042 (G.M.D), R24AA022057 (V.V.), R01AA028859 (Y.C.), AI150698 (J.M.), K01CA240533-01A1 (C.L.C.), and R01AR078000 (R.T.D.). Schematics in Figs. 1A, 2I, 4A, 5A, and 6A were created with BioRender.com.

## Author contributions

G.A. and I.S.H. initiated the study, conceived the project, designed experiments, interpreted results, and wrote the manuscript. G.A. performed the experiments with assistance from E.T.T. for animal breeding and experiments. N.P.W., Y.P.K., Y.K., and G.M.D. performed metabolite and lipidomic analyses. M.E.K. and B.F.C. performed proteomic analyses. N.G. performed immunofluorescent experiments. K.R., F.H., M.Z., L. S-S., T.Q.S., K.T., F.A., R.P.D., Z.R.S. assisted with animal necropsies. D.A-V., A.F.H., A.R.H., T.N.O., R.T.D., S.W., A.R., R.T.B., J.C., G.K.G., assisted with animal experiments and histological analyses. L.M.S. performed bioinformatic analyses. K.R. assisted with RNA analysis. H.C. and Z.S. assisted with immunoblot experiments. C.L.C. assisted with DEXA analyses. Y.C., V.V., D.B., X.L.S, and J.M. provided expert comments and reagents.

## Competing interests

I.S.H. reports financial support from Kojin Therapeutics. All other authors declare no competing interests.

## Additional information

¹Department of Biochemistry and Biophysics, University of Rochester Medical Center, Rochester, NY, USA. ²Department of Biomedical Genetics, University of Rochester Medical Center, Rochester, NY, USA. ³Wilmot Cancer Institute, University of Rochester Medical Center, Rochester, NY, USA. ⁴Department of Metabolism and Physiology, Moffitt Cancer Center and Research Institute, Tampa, FL, USA. ⁵Department of Chemistry and The Skaggs Institute for Chemical Biology, The Scripps Research Institute, La Jolla, CA, USA. ⁶Leiden Institute of Chemistry, Leiden University, Leiden, the Netherlands. ⁷Department of Cell Biology, Harvard Medical School, Boston, MA, USA. ⁸Department of Environmental Health Sciences, Yale School of Public Health, New Haven, CT, USA. ⁹Department of Pharmacology and Physiology, University of Rochester Medical Center, Rochester, NY, USA. ¹⁰Center for Musculoskeletal Research, University of Rochester Medical Center, Rochester, NY, USA. ¹¹Department of Pathology and Laboratory Medicine, University of Rochester Medical Center, Rochester, NY, USA. ¹²Leicester Cancer Research Centre, University of Leicester, Leicester, UK. ¹³Experimental and Molecular Immunology, Dept. of Infection and Immunity (DII), Luxembourg Institute of Health, Esch-sur-Alzette, Luxembourg. ¹⁴Immunology & Genetics, Luxembourg Centre for Systems Biomedicine (LCSB), University of Luxembourg, Esch-sur-Alzette, Luxembourg. ¹⁵Odense Research Center for Anaphylaxis (ORCA), Department of Dermatology and Allergy Center, Odense University Hospital, University of Southern Denmark, Odense, Denmark. ¹⁶Dipartimento di Bioscienze, Università degli Studi di Milano, Milan, Italy. ✉e-mail: isaac_harris@urmc.rochester.edu

