## [Peer Review File · Nature Communications]

Glutathione synthesis in the mouse liver supports lipid abundance through NRF2 repressionEditorial Note: Parts of this Peer Review File have been redacted as indicated to maintain the confidentiality of unpublished data.

REVIEWER COMMENTS

Reviewer #1 (Remarks to the Author):

To understand the requirement of GSH in adult tissue, the authors utilized mouse genetic approaches, inducible systemic disruption of Gclc gene, liver-specific Gclc knockout, and liver-specific double knockout of Gclc and Nrf2. The systemic Gclc disruption results in body weight loss and reduced white adipose tissue mass accompanied by NRF2 pathway activation and decreased expression of lipogenic genes. The liver-specific Gclc disruption by Cre-expressing viral infection induced similar changes in gene expression as well as in triglyceride levels in liver. The altered lipid metabolism was restored by simultaneous disruption of Nrf2. The authors concluded that GSH synthesis in liver is required for the maintenance of fat stores and lipid metabolism.

It is interesting to know that inducible Gclc inhibition in liver results in NRF2 activation and suppresses lipogenesis and reduces circulating lipid levels. Data presented are solid and the manuscript is well organized. However, because it has been well known that GSH depletion activates Nrf2 pathway, and that Nrf2 activation inhibits lipogenesis and promotes beta-oxidation, the conceptual advances could be rather incremental. Several comments and concerns need to be addressed to further strengthen the work.

1) Nrf2 protein accumulation by Gclc disruption needs to be shown by immunoblot analysis and immunohistochemistry.

2) Whole-animal Gclc knockout needs to be rescued by liver-specific expression of Gclc to argue that “liver-specific GSH synthesis promotes lipid abundance”.

3) Liver-specific Gclc knockout needs to be rescued by GSH supplementation to exclude the requirement of non-enzymatic function of Gclc.

4) Is Nrf2 activation in liver sufficient for the alterations in the lipid metabolism and abundance? Does liver-specific Keap1 disruption mimic liver-specific Gclc disruption? Or Gclc inhibition and Nrf2 activation are both needed for the phenotype?

4) How does Nrf2 suppress lipogenic genes? Molecular mechanisms would be shown.

Reviewer #2 (Remarks to the Author):

Asantewaa G et al examined the role of GSH synthesis in the liver of adult mice using animal models with induced global or liver-specific deletion of GCLC. They reported a dispensable role of GSH synthesis in adult liver survival. However, through a series of omics analyses (lipidomic, proteomic and metabolomics), they demonstrated a major function of liver GSH synthesis in sustaining lipid synthesis, circulating TGs and adipose tissue homeostasis. Mechanistically, they identified GSH deficiency may activate NRF2 to suppresses liver lipogenic gene expression. Although the research topic and some of the authors' discoveries are interesting, many data are very descriptive and provide no mechanistic insights into GSH role in adult liver. Some of the data is not solid enough to support their conclusions. Some of the major issues that need to be addressed are listed below.

Throughout the manuscript, the authors failed to demonstrate NRF2 expression at the protein level in GSH deficiency, which is essential as NRF2 is dominantly regulated at the posttranslational level by oxidative stress.

Oxidative stress activates NRF2/NQO1 signaling pathway which is known to be protective in pathological conditions by improving redox balance. Do Gclc KO and LKO livers develop primary oxidative stress, such as increased lipid peroxidation and superoxide and hydrogen peroxide? The pathological changes in liver, especially Gclc LKO, needs to be characterized in more details. The ALT is high in Gclc LKO mice. Is it associated with hepatocyte apoptosis or necroptosis? If ROS is activated in Gclc LKO liver, how about inflammation and fibrosis? More detailed pathological analyses of liver damages are needed along with molecular markers (other than only showing histology at low magnifications in most cases, such as Fig. S6).

NRF1 is also critical to the oxidative stress response in the adult liver, and it plays an

important role in oxidative stress-induced liver disease (PMID: 15738389). How about NRF1 expression in the liver of Gclc-LKO, Nrf2-LKO and L-DKO? Whether the dispensable role of NRF2 in the absence of GSH is due to the compensation of NRF1 needs to be investigated. Glutathione deficiency is associated with lipid synthesis; therefore, the whole concept of this manuscript is not novel. NRF2 has pleiotropic effects on lipogenesis, such as by regulating Srebp1 or FXR transcription. Liver-specific NRF2 has been shown to prevent high fat diet-induced NAFLD by dampening the expression of PPAR γ and its downstream lipogenic genes (PMID: 31901728). However, current manuscript did not provide any direct evidence linking GSH deficiency and NRF2 activation to any of these or novel mechanisms. Further mechanistic studies are needed.

Thus far, there is no sufficient evidence supporting the claim that “the repression of lipogenic gene expression and serum triglyceride deficiency in Gclc LKO mice is NRF2 dependent”. The rescuing effects of NRF2 deficiency in Gclc deficiency are minimal, arguing against NRF2 as the major target of GSH deficiency. A more comprehensive examination of other lipogenic genes identified by RNA-Seq and proteomics (Fig. 2K-L) in addition to Scd1 are necessary to provide more robust evidence of the role of NRF2 in regulating lipogenic genes.

How about the lipogenic gene expression and VLDL secretion in isolated primary hepatocytes from Gclc LKO and DLKO mice?

Both Gclc KO and Gclc-LKO mice demonstrate loss of adipose tissue, however, the underlying mechanisms were not interrogated. Especially, how do Gclc-LKO mice cause adipose fat loss? What about the GSH levels and oxidative stress in the adipose tissue of Gclc LKO mice? The pathophysiology and the underlying mechanisms of adipose tissue phenotype in Gclc-LKO mice need to be characterized in more details, including histology, molecular changes (inflammation, lipolysis or hypertrophy etc) and whole animal metabolic homeostasis. Are they different from the phenotype observed in eWAT of Gclc KO mice (Fig. 1K)? Does hepatic NRF2 deficiency in Gclc LKO mice also rescue adipose phenotype? These are important questions that need to be answered to ensure the thoroughness of the study. It would be interesting to explore which tissue contributes to the lethal phenotype in adult animals with deficiency in GSH synthesis?

Despite extensive omics data, it is hard to differentiate whether these changes are the primary and secondary effects of the GSH deficiency, especially in mice with induced global

deletion of Gclc. In both Gclc KO and LKO mice, how about the serum FFA and glucose levels? How about liver VLDL secretion? These factors contribute to the serum and liver TG levels. Meanwhile, the descriptions on serum lipid changes in Gclc LKO mice are very vague and incomplete considering the massive changes in various lipid species other than triglycerides (Fig. 5F).

Fig. 1L: List how many independent animals for each genotype are quantified for adipocyte sizes.

Fig. 2E and Fig. 2K, animal numbers for proteomic analyses are not consistent, n=3 or 6?

Fig. 2E, 2K, Fig. 3 A-C: data are not plotted based on Log2 fold change >2. It seems to be FC >2. Please correct.

Fig. 4D and Fig. 5H, if there are no differences in survival, no figures are necessary to show as one can hardly differentiate different genotypes.

The number of cohort animals for some quantitative analyses is too low. For example, Fig. S4: n=2 or 4.

Animal sex is not provided consistently in most experiments. Whether animals were fasted or not before harvesting serum or tissues were not clear. This is important since it affects interpretation of the crosstalk between liver, serum and adipose lipid profiles.

Fig. 4K include TG levels in WT under normal chow diet. Fig. S4K: indicate how many weeks after TAM.

Reviewer #3 (Remarks to the Author):

This manuscript addresses the effect of profound GSH depletion induced by GCLC deletion on the Nrf2 repression of hepatic triglyceride export leading to lethal weight loss.

1- The condition of such profound sustained hepatic GSH depletion is almost never achieved in any circumstance so not clear what is the relevance of the findings.

2- The issue of dissociating GSH depletion and Nrf2 is important and only supported by DKO. Other substantiation is needed – e.g. alternative ways of Nrf2 activation in absence of GSH depletion, e.g. overexpression of Nrf2 or chemical activation of Nrf2. This could be explored in cell culture models.

3- GSH turnover in the liver is mainly determined by the release into blood and bile leading to maintenance of steady levels of cysteine systemically (Seminars Liver Disease 18:313-329,

1998). Therefore, sustained marked depletion of hepatic GSH will be expected to impair the interorgan homeostasis. What happened to the levels of GSH in extrahepatic organs when hepatic GCLC was deleted?

4- Some insight into the extent of GSH depletion and/or Nrf2 activation required to impair triglyceride metabolism in the liver would enhance the manuscript.

We appreciate the Reviewers' comments and suggestions. The revisions have significantly strengthened the evidence supporting the manuscript's conclusions. **In total, we have added 60 new Figure panels and 11 additional Supplemental Figures.**

Specific responses to Reviewer comments:

Reviewer comments – Red

Author responses – Black

Text from the manuscript – Bolded in Black

Reviewer #1 (Remarks to the Author):

To understand the requirement of GSH in adult tissue, the authors utilized mouse genetic approaches, inducible systemic disruption of Gclc gene, liver-specific Gclc knockout, and liver-specific double knockout of Gclc and Nrf2. The systemic Gclc disruption results in body weight loss and reduced white adipose tissue mass accompanied by NRF2 pathway activation and decreased expression of lipogenic genes. The liver-specific Gclc disruption by Cre-expressing viral infection induced similar changes in gene expression as well as in triglyceride levels in liver. The altered lipid metabolism was restored by simultaneous disruption of Nrf2. The authors concluded that GSH synthesis in liver is required for the maintenance of fat stores and lipid metabolism.

It is interesting to know that inducible Gclc inhibition in liver results in NRF2 activation and suppresses lipogenesis and reduces circulating lipid levels. Data presented are solid and the manuscript is well organized. However, because it has been well known that GSH depletion activates Nrf2 pathway, and that Nrf2 activation inhibits lipogenesis and promotes beta-oxidation, the conceptual advances could be rather incremental. Several comments and concerns need to be addressed to further strengthen the work.

We appreciate the kind words from Reviewer #1 regarding the rigor of the data and the organization of the manuscript. We also appreciate that the conceptual advances could be viewed as incremental. To address this, we have added additional findings to our manuscript, specifically that the loss of liver-specific GCLC induces a sex-dependent requirement for NRF2 to maintain liver homeostasis and animal survival. We have included these findings in Figure 6 (shown below) and Supplementary Figures S15, S16, and S17.

[figure redacted]

1) Nrf2 protein accumulation by Gclc disruption needs to be shown by immunoblot analysis and immunohistochemistry.

We appreciate this important point by the Reviewer. We now demonstrate with immunoblot analysis that NRF2 protein accumulates in the liver of liver-specific Gclc KO mice as compared to Gclc WT mice (**Fig. 4E**). Further, we demonstrate that the accumulation of NRF2 protein does not occur in the liver of liver-specific Gclc-Nrf2 DKO mice (**Fig. 5E**), demonstrating the specificity of the antibody used to identify NRF2 protein.

[figure redacted]

[figure redacted]

We tried to measure NRF2 protein using IHC approaches with a publicly available NRF2 antibody (i.e., Abcam ab31163) but did not observe any differences in liver-specific Gclc KO liver tissue. We hypothesize that the lack of signal is due to the low sensitivity of the NRF2 antibodies.

2) Whole-animal Gclc knockout needs to be rescued by liver-specific expression of Gclc to argue that “liver-specific GSH synthesis promotes lipid abundance”.

Whole-animal Gclc KO mice rapidly lose body weight, have reductions in fat mass and lean mass, and have reduced levels of circulating triglycerides. Liver-specific Gclc KO mice show reduced levels of circulating triglycerides but do not rapidly lose body weight or have reductions in fat mass and lean mass. We hypothesized that the loss of body weight and reductions in lean mass and fat tissue were caused by the loss of Gclc in a tissue other than the liver. Further analysis revealed that whole-animal Gclc KO mice show intestinal damage, including enteritis and colitis (**Fig. R1**). Additionally, we find that intestinal-specific Gclc KO mice rapidly lose body weight and have colitis but do not show reduced levels of circulating triglycerides, suggesting that loss of GSH in the intestine is responsible for the rapid loss of body weight and fat and lean mass in whole-animal Gclc KO mice. We have provided this data for Reviewers. For clarity, we have removed the data regarding reductions in fat mass and lean mass in whole-animal Gclc KO mice. Additionally, since these phenotypes are distinct from the liver, circulating triglycerides, and lipogenesis, we are requesting to exclude this data from the manuscript so that we can fully explore these phenotypes in future studies.

[figure redacted]

3) Liver-specific Gclc knockout needs to be rescued by GSH supplementation to exclude the requirement of non-enzymatic function of Gclc.

We appreciate the Reviewers' point regarding the need to rescue the phenotypes from liver-specific Gclc KO mice with GSH supplementation to exclude any potential non-enzymatic functions of GCLC being responsible. This was technically challenging to test since supplemented GSH is reported not to be directly taken up by cells but instead broken down and re-synthesized inside the cell by Gclc (Griffith OW, Proc Natl Acad Sci U S A, 1978, PMID: 31622). GSH ethyl-ester (GSHee), which readily enters cells (Levy EJ, Proc Natl Acad Sci U S A, 1993, PMID: 8415673), has been used in previous studies at doses ranging from 4 mg/kg to over 3000 mg/kg (PMIDs: 4004275, 33313264, and 11087910). We injected Gclc^{ff} mice with AAV-TBG-Cre (to induce liver-specific Gclc deletion) and delivered GSH-ee to mice at a range of doses (0, 50, 100, 500, 1000, 2000 mg/kg) during week 3 from days 17 to 20 daily. This time point was chosen because this is where the highest drop in GSH and induction in NRF2 target genes was observed. Further, mice receiving injections of 2000 mg/kg had reduced survival (data not shown). Interestingly, we observed a dose-dependent reduction in NRF2 target genes Nqo1 and Gclm with increasing concentrations of GSH-ee (**Fig. 4J and S9A**). However, we could not observe an increase in lipogenic genes Scd1 and Fasn (**Fig. 4K and S9B**) or an increase in serum triglyceride levels (**Fig. 4L**) with the increasing concentrations of GSH-ee. We hypothesize that the lack of rescue in lipogenic gene expression and serum triglycerides in liver-specific Gclc KO treated with

GSH-ee may partly be due to the technical limitations of fully restoring GSH levels in the mice. We have shown the data below and now include the following statement in the manuscript:

“These data indicate that GSH depletion is directly responsible for the observed induction of Nrf2 activity in Gclc L-KO mice. However, while technical limitations may exist regarding adequate GSH rescue, we cannot rule out non-canonical and non-enzymatic functions of GCLC contributing to lipid-related phenotypes.”

[figure redacted]

[figure redacted]

4) Is Nrf2 activation in liver sufficient for the alterations in the lipid metabolism and abundance? Does liver-specific Keap1 disruption mimic liver-specific Gclc disruption? Or Gclc inhibition and Nrf2 activation are both needed for the phenotype?

We appreciate this question from the Reviewer. Several groups have studied Nrf2 activation through disruption of KEAP1 using both genetic and pharmacologic approaches. A Keap1 hypomorph mouse strain was shown to repress the expression of lipid genes and dampen weight gain following a high-fat diet (Slocum SL, Arch Biochem Biophys, 2016, PMID: 26701603). Similarly, a liver-specific Keap1 deletion was shown to lower the expression of lipid genes and reduce liver steatosis induced by a Western diet (Ramadori P, Free Radic Biol Med., 2016, PMID: 26698665). Pharmacologically, KEAP1 activity can be inhibited by bardoxylone (CDDO). Treatment of mice with CDDO-imidazolide (CDDO-im) lowered the expression of lipid genes and dampened body weight gain caused by a high-fat diet (Shin S, Eur J Pharmacol, 2009, PMID: 19698707). Additionally, the treatment of mice with the NRF2-inducing compound TBE-31 suppressed the expression of lipogenic genes and lowered liver steatosis in mice following a high-fat, high-fructose diet (Sharma RS, Cell Mol Gastroenterol Hepatol, 2017, PMID: 29552625).

We tested how direct NRF2 activation using bardoxylone-methyl (CDDO-me) compares to NRF2 activation following liver-specific Gclc deletion. Notably, while NRF2 protein accumulation could be observed in whole-protein lysate of liver tissue from liver-specific Gclc KO mice (**Fig. 4E and 5E**), this was barely visible even under high exposure in liver tissue from mice treated with CDDO-me (**Fig. S14A**). Following the enrichment of nuclear lysate, NRF2 protein could be detected in liver tissue from mice treated with CDDO-me (**Fig. S14B**); however, these levels were still not as

elevated compared to those observed in liver-specific Gclc deletion mice. We found that the treatment with CDDO-me caused an induction of NRF2 target genes (i.e. Nqo1), repression of lipogenic enzymes (i.e. Scd1), and lower levels of serum triglycerides (**Figure 5J-5L and Figure S14C-S14E**). In line with the muted accumulation of NRF2 protein, CDDO-me induced NRF2 targets and repressed lipogenic genes to a dramatically lower degree compared to liver-specific Gclc disruption. We hypothesize that we observed a greater induction of NRF2 through liver-specific Gclc deletion (as compared to KEAP1 inhibition with CDDO-me) because GSH prevents NRF2 activation through KEAP1-independent mechanisms. Indeed, it was recently reported that treatment of cells with H₂O₂, which can lower GSH levels, leads to increased expression of NRF2 target genes in a Keap1-independent mechanism (Al-Mubarak BR, Redox Biol, 2021, PMID: 34626892). We have shown the data below and now include the following statement in the manuscript:

[figure redacted]

[figure redacted]

“Our results indicate that GSH supports lipid abundance by preventing NRF2 activation in the liver, as well as through other potential NRF2-independent mechanisms. NRF2 activation has been shown to repress the expression of lipogenic genes¹⁻⁴. NRF2 protein stability is regulated by KEAP1. Pharmacologically, KEAP1 activity can be inhibited by the compound bardoxolone (CDDO)⁵. Therefore, to further dissociate the effects of GSH and NRF2 on lipid abundance and assess if NRF2 activation alone mimics the effects of liver-specific Gclc deletion, we treated Gclc WT mice with CDDO-methyl (CDDO-Me), a form of bardoxolone with increased bioavailability. Treatment with CDDO-Me increased NRF2 expression but not to the level observed in the liver of Gclc L-KO mice, as we could only detect substantial NRF2 expression after enriching the whole protein lysate for a nuclear fraction (Figure S14A-S14B). CDDO-Me treatment increased the expression of NRF2 target genes (Figure S14C-S14D) and lowered the expression of lipogenic genes (Figure 5K, S14E). Consequently, CDDO-Me treatment caused a reduction in serum triglycerides (Figure 5L). However, these resulting effects of NRF2 induction via CDDO treatment were to a dramatically lower degree compared to the effects of NRF2 induction upon GSH depletion in Gclc L-KO mice (Figure 4E-4F and 5D-5E). CDDO-Methyl treatment did not impact body weight, liver damage, or serum glucose and cholesterol levels (Figure S14F-S14J). We hypothesize that we observed a greater induction of NRF2 through liver-specific Gclc deletion (as compared to KEAP1 inhibition with CDDO-me) because GSH prevents NRF2 activation through KEAP1-independent mechanisms. Indeed, it was recently reported that treatment of cells with H₂O₂, which can lower GSH levels, leads to increased expression of NRF2 target genes in a Keap1-independent mechanism⁶.”

5) How does Nrf2 suppress lipogenic genes? Molecular mechanisms would be shown.

We appreciate the question regarding the molecular mechanisms by which Nrf2 suppresses lipogenic gene expression following GSH depletion. Previously, it has been suggested that Nrf2 suppresses lipogenic genes by inhibiting LXR activity, a transcription factor that supports lipogenic gene expression (Kay HY, Antioxid Redox Signal, 2011, PMID: 21504366). We hypothesized that a similar mechanism occurs following GSH depletion in the liver. To test this, we treated control and liver-specific Gclc KO mice with the LXR agonist T0901317 (i.p. 50 mg/kg, daily) and for the duration of GSH depletion (i.e., days 7-21 following injection with AAV-TBG-Cre). We found that induction of LXR activity with T0901317 rescued lipogenic gene expression and serum triglycerides in liver-specific Gclc KO mice (**Figure 5M-5N**). These data suggest that NRF2 activation following GSH depletion causes repression of LXR activity, leading to lower expression of lipogenic enzymes and lower levels of serum triglycerides. We have shown the data below and now include the following statement in the manuscript:

[figure redacted]

“NRF2 is reported to have suppressive effects on the lipogenic pathway via inhibiting the activity of LXR^{1,7}, a transcription factor that supports lipid synthesis. To test if this mechanism was at play upon depleting GSH in the liver, we treated Gclc L-KO mice with T0901317, an LXR agonist. Induction of LXR activity in the Gclc L-KO mice rescued lipogenic gene expression and circulating triglycerides (Figure 5M-5N). These data suggest that NRF2 activation following GSH depletion causes repression of LXR activity, leading to lower expression of lipogenic enzymes and lower levels of serum triglycerides.”

Reviewer #2 (Remarks to the Author):

Asantewaa G et al examined the role of GSH synthesis in the liver of adult mice using animal models with induced global or liver-specific deletion of GCLC. They reported a dispensable role of GSH synthesis in adult liver survival. However, through a series of omics analyses (lipidomic, proteomic and metabolomics), they demonstrated a major function of liver GSH synthesis in sustaining lipid synthesis, circulating TGs and adipose tissue homeostasis. Mechanistically, they identified GSH deficiency may activate NRF2 to suppresses liver lipogenic gene expression. Although the research topic and some of the authors' discoveries are interesting, many data are very descriptive and provide no mechanistic insights into GSH role in adult liver. Some of the data is not solid enough to support their conclusions. Some of the major issues that need to be addressed are listed below.

We appreciate that the Reviewer finds some of the discoveries interesting and has provided important feedback to improve the manuscript.

Throughout the manuscript, the authors failed to demonstrate NRF2 expression at the protein level in GSH deficiency, which is essential as NRF2 is dominantly regulated at the posttranslational level by oxidative stress.

We appreciate this important point by the Reviewer. We now demonstrate with immunoblot analysis that NRF2 protein accumulates in the liver of liver-specific Gclc KO mice as compared to Gclc WT mice (**Fig. 4E**). Further, we demonstrate that the accumulation of NRF2 protein does not occur in the liver of liver-specific Gclc-Nrf2 DKO mice (**Fig. 5E**), demonstrating the specificity of the antibody used to identify NRF2 protein.

[figure redacted]

[figure redacted]

Oxidative stress activates NRF2/NQO1 signaling pathway which is known to be protective in pathological conditions by improving redox balance. Do Gclc KO and LKO livers develop primary oxidative stress, such as increased lipid peroxidation and superoxide and hydrogen peroxide?

We appreciate this question from the Reviewer. We have performed immunoblot analysis by staining for 8-oxoguanine, a marker of oxidized DNA, and 4-hydroxynoneal (4-HNE), a marker of lipid peroxidation, and found no changes in levels in the livers of liver-specific Gclc KO mice compared to control mice at 3 weeks following induction of the deletion with AAV-TBG-Cre (**Fig. S6B and S13A-S13B**). However, we did observe increased staining for 8-oxoguanine in the livers of liver-specific Gclc KO mice compared to control mice at the extended timepoint of 10 weeks (**Fig. S16**). We did not measure superoxide and hydrogen peroxide in tissues using immunohistochemistry, as it is not recommended to use probes for these species for immunohistochemical analysis (Murphy MP, Nat Metab, 2022, PMID: 35760871). We have shown the data below and now include the following statement in the manuscript:

[figure redacted]

“Notably, no significant increases in markers of inflammation or oxidative stress were observed in the liver of *Gclc* L-KO mice compared to *Gclc* WT mice (Figure S6B). These data demonstrate that deletion of the GCLC protein can be induced in the liver of adult animals. Importantly, these phenotypes contrast with the non-inducible liver-specific *Gclc* KO mouse strain (*Gclc*^{ff} Albumin-Cre), which develops liver failure and dies shortly after birth⁸. “

[figure redacted]

“Additionally, no changes in markers of oxidative DNA damage (Figure S13A) and lipid peroxidation (Figure S13B) were observed in the L-DKO mice. These findings further suggest that at acute time points, both GCLC and NRF2 are dispensable to the liver.”

[figure redacted]

“Further analysis revealed that male L-DKO livers had dramatically increased markers of apoptosis and oxidative DNA damage compared to female mice (Figure S16).

The pathological changes in liver, especially Gclc LKO, needs to be characterized in more details. The ALT is high in Gclc LKO mice. Is it associated with hepatocyte apoptosis or necroptosis? If ROS is activated in Gclc LKO liver, how about inflammation and fibrosis? More detailed pathological analyses of liver damages are needed along with molecular markers (other than only showing histology at low magnifications in most cases, such as Fig. S6).

We appreciate the Reviewers' questions regarding the characterization of liver tissue from liver-specific Gclc KO mice. To measure apoptosis, we performed immunoblot and immunohistochemistry staining for cleaved caspase 3 and found no changes in the liver from liver-specific Gclc KO mice compared to control mice (**Fig. S6A**). Additionally, we performed TUNEL staining and found no changes in the liver of liver-specific Gclc KO mice (**Fig. S6A**). To measure necroptosis, we performed immunohistochemistry against phosphorylated MLKL and found no differences in the livers of liver-specific Gclc KO mice (**Fig. S6A**). We, however, observed increased markers of apoptosis in liver-specific Gclc KO male mice at the extended time point of 10 weeks (**Fig. S16**). To measure inflammation and fibrosis, we performed immunohistochemical staining for F4/80, which is a marker of macrophages, YAP/TAZ, which are markers of fibrosis (Akl MG, Cell Rep, 2023, PMID: 37060561), and p62, which is a marker of autophagy. We observed no difference in levels of F4/80 staining in liver-specific Gclc KO mice (**Fig. S6B**). Interestingly, we observed increased staining for YAP/TAZ and p62, which was largely reversed in liver-specific Gclc-Nrf2 DKO mice (**Fig. S12B-S12C**). These data suggest that while markers of fibrosis and autophagy are increased following liver-specific Gclc deletion, these are potentially due to the activation of Nrf2. Finally, histological slides were scored for damage by board-certified clinical pathologists (Dr. Aaron Huber and Dr. Diana Agostini-Vulaj). We observed slight but significant increases in hepatocellular injury scores in the livers of liver-specific Gclc KO mice (**Fig. S5E**). Similar trends were observed in liver-specific Gclc KO mice at the extended timepoint of 10 weeks (**Fig. 6D**). We concluded that a level of liver damage occurred; however, this level of damage was dramatically less than that observed in mice with a liver-specific Gclc KO driven by albumin-Cre (Alb-Cre) (PMID: 17464988). We have shown the data below and now include the following statement in the manuscript:

[figure redacted]

“Further histopathological analyses showed that the elevation in liver damage biomarkers observed in the Gclc L-KO mice was not associated with increased markers of apoptosis but was associated with increased markers of necroptosis (Figure S6A).”

[figure redacted]

“Further analysis revealed that male L-DKO livers had dramatically increased markers of apoptosis and oxidative DNA damage compared to female mice (Figure S16).”

[figure redacted]

“Notably, no significant increases in markers of inflammation or oxidative stress were observed in the liver of Gclc L-KO mice compared to Gclc WT mice (Figure S6B).”

[figure redacted]

“Pro-fibrotic markers yes-associated protein 1 (YAP1) and taffazin (TAZ) and autophagy marker p62 showed elevated expression in Gclc L-KO mice (Figure S12B-S12C).

Interestingly, these markers largely reversed in liver-specific Gclc-Nrf2 DKO mice. These data suggest that while markers of fibrosis and autophagy are increased following liver-specific Gclc deletion, these are potentially due to the activation of Nrf2”

[figure redacted]

“analysis of hepatocellular injury and serum AST and ALT levels showed a significant increase compared to Gclc WT mice (Figure S5E-S5G).”

[figure redacted]

“histological analysis revealed a significant difference in liver damage between female and male L-DKO mice (Figure 6D)”

NRF1 is also critical to the oxidative stress response in the adult liver, and it plays an important role in oxidative stress-induced liver disease (PMID: 15738389). How about NRF1 expression in the liver of Gclc-LKO, Nrf2-LKO and L-DKO? Whether the dispensable role of NRF2 in the absence of GSH is due to the compensation of NRF1 needs to be investigated.

We appreciate the Reviewers' question regarding NRF1 expression in the liver of Gclc L-KO, Nrf2 L-KO, and L-DKO mice. We performed immunoblot analysis for NRF1 protein and found NRF1 levels were increased in the liver of Gclc L-KO mice (**Fig. S12A**). Interestingly, the levels of NRF1 in the livers of L-DKO mice were lower than in the Gclc L-KO mice; however, they remained higher than control mice. These data suggest that GSH synthesis, potentially through NRF2 repression, also limits NRF1 protein expression in the liver. Further, these data suggest that at acute time points following Gclc genetic deletion, compensation by NRF1 could be responsible for the dispensable role of NRF2 in GSH-depleted conditions.

Importantly, we have examined Gclc-Nrf2 L-DKO at an extended time point of 10 weeks and found that males (but not females) have reduced survival (**Fig. 6 and Fig. S15, S16, and S17**). This suggests that female mice can compensate for the absence of NRF2 under GSH depletion as compared to males.

We have shown the data below and now include the following statement in the manuscript:

[figure redacted]

“NRF1 is a transcription factor that belongs to the same transcription factor family as NRF2. While NRF1 and NRF2 are differentially regulated by KEAP1⁹, both have overlapping targets and roles in preventing liver-related diseases¹⁰. Further, NRF1 has been shown to be important for the survival of hepatocytes and to mediate oxidative stress during development^{11,12}. Thus, we examined if the observed dispensable role of NRF2 in the absence of GSH was due to a compensatory activity by NRF1. We observed that livers from Gclc L-KO mice had increased expression of NRF1 protein (Figure S12A). Interestingly, the levels of NRF1 in the livers of L-DKO mice were lower than in the Gclc L-KO mice; however, they remained higher than control mice. These data suggest that GSH synthesis, potentially through NRF2 repression, also limits NRF1 protein expression in the

liver. Further, these data suggest that at acute time points following *Gclc* genetic deletion, compensation by NRF1 could be responsible for the dispensable role of NRF2 in GSH-depleted conditions. Further research is required to fully elucidate the interplay of NRF1 and NRF2 following GSH depletion in the liver.”

[figure redacted]

[figure redacted]

[figure redacted]

[figure redacted]

“To test the effect of sustained repression of lipogenic programs and a concurrent sustained induction of NRF2 activity, we monitored mice for ten weeks following the induction of *Gclc* deletion alone (*Gclc* L-KO) or in combination with *Nrf2* deletion (L-DKO) in the liver (Figure 6A). Nearly all liver-specific *Gclc* L-KO mice survived over time (Figure 6B). However, we found the L-DKO mice had a reduced survival rate which was sex-specific (Figure 6B-6C). Male mice underwent rapid loss of survival at extended time points, while female mice were largely unaffected. In addition, histological analysis revealed a significant difference in liver damage between female and male L-DKO mice (Figure 6D and 6E). Indeed, we found male L-DKO mice to have greater hepatocellular injury than female *Gclc* L-KO mice (Figure 6D and 6E). Liver damage markers in the serum were also found to be significantly elevated in the serum of male L-DKO mice; however, they were not greater than levels from male *Gclc* L-KO mice (Figure S15A-S15B). Importantly, these levels were still lower than those reported in the non-inducible liver-specific *Gclc* KO mouse strain⁸. Like the acute time points of liver-specific *Gclc* deletion, serum triglycerides were depleted in mice with prolonged liver-specific *Gclc* deletion and rescued with *Nrf2* deletion (Figure 6F). This was found to not be sex-dependent, as the rescue was observed in both the female and male mice. (Figure S15C). Unlike the acute setting, however, prolonged ablation of GSH synthesis in the liver resulted in decreased abundance of epididymal white adipose depots, a major storage tissue for triglycerides (Figure 6G). This was predominant in the male *Gclc* L-KO mice and was not rescued with *Nrf2* deletion (Figure S15D). Additionally, we did not find any significant changes in serum levels of cholesterol and glucose, which might contribute to serum triglycerides (Figure S15E – S15F). Further analysis revealed that male L-DKO livers had dramatically increased markers of apoptosis and oxidative DNA damage compared to female mice (Figure S16). Interestingly, we found increased apoptotic markers in the adipose tissue of both female and male L-DKO mice compared to L-*Gclc* KO mice (Figure S17). However, more research is required to fully elucidate the mechanisms involved in the adipose tissue of these mice. Together, these data reveal a sex-dependent requirement for NRF2 in the absence of GSH over time. Further, these data suggest that by maintaining circulating triglycerides, GSH synthesis in the liver potentially supports fat stores in the body.”

Glutathione deficiency is associated with lipid synthesis; therefore, the whole concept of this manuscript is not novel.

We appreciate that the conceptual advances could be viewed as not novel. To address this, we have added additional findings to our manuscript, specifically that the loss of liver-specific GCLC induces a sex-dependent requirement for NRF2 at an extended time point of 10 weeks (**Fig. 6 and Fig. S15, S16, and S17**).

NRF2 has pleiotropic effects on lipogenesis, such as by regulating *Srebp1* or FXR transcription. Liver-specific NRF2 has been shown to prevent high fat diet-induced NAFLD by dampening the expression of PPAR γ and its downstream lipogenic genes (PMID: 31901728). However, current manuscript did not provide any direct evidence linking GSH deficiency and NRF2 activation to any of these or novel mechanisms. Further mechanistic studies are needed.

We appreciate the question regarding the molecular mechanisms by which Nrf2 suppresses lipogenic gene expression following GSH depletion. Previously, it has been suggested that Nrf2 suppresses lipogenic genes by inhibiting LXR activity, a transcription factor that supports lipogenic gene expression (Kay HY, Antioxid Redox Signal, 2011, PMID: 21504366). We hypothesized that a similar mechanism occurs following GSH depletion in the liver. To test this, we treated control and liver-specific Gclc KO mice with the LXR agonist T0901317 (i.p. 50 mg/kg, daily) and for the duration of GSH depletion (i.e., days 7-21 following injection with AAV-TBG-Cre). We found that induction of LXR activity with T0901317 rescued lipogenic gene expression and serum triglycerides in liver-specific Gclc KO mice (**Figure 5M-5N**). These data suggest that NRF2 activation following GSH depletion causes repression of LXR activity, leading to lower expression of lipogenic enzymes and lower levels of serum triglycerides. We have shown the data below and now include the following statement in the manuscript:

[figure redacted]

“NRF2 is reported to have suppressive effects on the lipogenic pathway via inhibiting the activity of LXR^{1,7}, a transcription factor that supports lipid synthesis. To test if this mechanism was at play upon depleting GSH in the liver, we treated Gclc L-KO mice with T0901317, an LXR agonist. Induction of LXR activity in the Gclc L-KO mice rescued lipogenic gene expression and circulating triglycerides (Figure 5M-5N). These data suggest that NRF2 activation following GSH depletion causes repression of LXR activity, leading to lower expression of lipogenic enzymes and lower levels of serum triglycerides.”

Thus far, there is no sufficient evidence supporting the claim that “the repression of lipogenic gene expression and serum triglyceride deficiency in Gclc LKO mice is NRF2 dependent”. The rescuing effects of NRF2 deficiency in Gclc deficiency are minimal, arguing against NRF2 as the major target of GSH deficiency. A more comprehensive examination of other lipogenic genes identified by RNA-Seq and proteomics (Fig. 2K-L) in addition to Scd1 are necessary to provide more robust evidence of the role of NRF2 in regulating lipogenic genes.

We appreciate the Reviewers’ question regarding other lipogenic genes controlled by NRF2 following GSH depletion in the liver. We performed RNA-seq and proteomics on livers of control, Gclc L-KO, Nrf2 L-KO, and Gclc-Nrf2 L-DKO mice (**Fig. 5H-5I and Supplemental Table S5**). Using gene lists from GSEA pathways identified to be downregulated in RNA in the liver of Gclc L-KO mice, we found several lipogenic factors that are downregulated at the RNA and protein levels in the liver of Gclc L-KO mice are reversed in the liver of Gclc/Nrf2 L-DKO mice. We have shown the data below and now include the following statement in the manuscript:

[figure redacted]

“Notably, the repression of lipogenic gene and protein expression and the decreased serum triglyceride levels seen in Gclc L-KO mice were reversed in L-DKO mice (Figure 5F-5J, and Table S5). For certain lipogenic genes, however, the reversal of downregulated expression was not complete (Figure 5F), suggesting the potential involvement of NRF2-independent pathways. Together, these results indicate that GSH supports lipid abundance by preventing NRF2 activation in the liver.”

How about the lipogenic gene expression and VLDL secretion in isolated primary hepatocytes from Gclc LKO and DLKO mice?

We appreciate the suggestion to examine isolated primary hepatocytes from Gclc LKO and DLKO mice. The phenotypes that we describe in the manuscript focus on the impact of liver-specific Gclc KO on serum triglycerides in vivo, and we believe that investigating lipogenic gene expression and VLDL secretion in isolated primary hepatocytes is beyond the scope of this manuscript.

Both Gclc KO and Gclc-LKO mice demonstrate loss of adipose tissue, however, the underlying mechanisms were not interrogated. Especially, how do Gclc-LKO mice cause adipose fat loss? What about the GSH levels and oxidative stress in the adipose tissue of Gclc LKO mice? The pathophysiology and the underlying mechanisms of adipose tissue phenotype in Gclc-LKO mice need to be characterized in more details, including histology, molecular changes (inflammation, lipolysis or hypertrophy etc) and whole animal metabolic homeostasis. Are they different from the phenotype observed in eWAT of Gclc KO mice (Fig. 1K)? Does hepatic NRF2 deficiency in Gclc LKO mice also rescue adipose phenotype? These are important questions that need to be answered to ensure the thoroughness of the study.

We have removed the data pertaining to adipose tissue loss in the Gclc KO mice and instead only show the phenotypes observed in eWAT in liver-specific Gclc KO mice at extended time periods (i.e., 10 weeks). We tested whether the decrease in eWAT is rescued in Gclc/Nrf2 L-DKO mice. Interestingly, we found that, unlike in the acute setting, at extended time points, NRF2 is required in the absence of GCLC in the liver in a sex-dependent manner (**Figure 6B-6C**). We found that male mice undergo rapid loss of survival at extended time points, while female mice are largely unaffected. Interestingly, Gclc/Nrf2 L-DKO female mice have reduced eWAT, arguing that this impact on eWAT by GSH deficiency at extended time points is not dependent on NRF2 activation.

We have included these findings in the manuscript and now state:

[figure redacted]

[figure redacted]

Unlike the acute setting, however, prolonged ablation of GSH synthesis in the liver resulted in decreased abundance of epididymal white adipose depots, a major storage tissue for triglycerides (Figure 6G). This was predominant in the male *Gclc* L-KO mice and was not rescued with *Nrf2* deletion (Figure S15D).

[figure redacted]

Interestingly, we found increased apoptotic markers in the adipose tissue of both female and male L-DKO mice compared to L-Gclc KO mice; however, more research is required to fully elucidate the mechanisms involved in the adipose tissue of these mice (Figure S17).

It would be interesting to explore which tissue contributes to the lethal phenotype in adult animals with deficiency in GSH synthesis?

Whole-animal Gclc KO mice rapidly lose body weight, have reductions in fat mass and lean mass, and have reduced levels of circulating triglycerides. Liver-specific Gclc KO mice show reduced levels of circulating triglycerides but do not rapidly lose body weight or have reductions in fat mass and lean mass. We hypothesized that the loss of body weight and reductions in lean mass and fat tissue were caused by the loss of Gclc in a tissue other than the liver. Further analysis revealed that whole-animal Gclc KO mice show intestinal damage, including enteritis and colitis (**Fig. R1**). Additionally, we find that intestinal-specific Gclc KO mice rapidly lose body weight and have colitis but do not show reduced levels of circulating triglycerides, suggesting that loss of GSH in the intestine is responsible for the rapid loss of body weight and fat and lean mass in whole-animal Gclc KO mice. We have provided this data for Reviewers. For clarity, we have removed the data regarding reductions in fat mass and lean mass in whole-animal Gclc KO mice. Additionally, since these phenotypes are distinct from the liver, circulating triglycerides, and lipogenesis, we are requesting to exclude this data from the manuscript so that we can fully explore these phenotypes in future studies.

[figure redacted]

Despite extensive omics data, it is hard to differentiate whether these changes are the primary and secondary effects of the GSH deficiency, especially in mice with induced global deletion of Gclc. In both Gclc KO and LKO mice, how about the serum FFA and glucose levels? How about liver VLDL secretion? These factors contribute to the serum and liver TG levels.

We agree with the Reviewer and now provide glucose levels and cholesterol levels from mice.

We have included these findings as Supplementary Figures S7 and S15 and now state:

[figure redacted]

Liver tissue from Gclc L-KO mice showed enrichment of NRF2 protein and downstream target genes (Figure 4D-4F, S7A-B, and Table S4). Like the whole-body Gclc KO mice, we observed a decrease in lipogenic factors and serum triglycerides over time (Figure 4G-4I, S7C, and Table S4). This was associated with changes in serum glucose levels but not serum cholesterol levels (Figure S7D-S7E).

[figure redacted]

Unlike the acute setting, however, prolonged ablation of GSH synthesis in the liver resulted in decreased abundance of epididymal white adipose depots, a major storage tissue for triglycerides (Figure 6G). This was predominant in the male Gclc L-KO mice and was not

rescued with *Nrf2* deletion (Figure S15D). Additionally, we found sex-dependent changes in serum levels of cholesterol and glucose, which might contribute to serum triglycerides (Figure S15E – S15F).

Meanwhile, the descriptions on serum lipid changes in *Gclc* LKO mice are very vague and incomplete considering the massive changes in various lipid species other than triglycerides (Fig. 5F).

We appreciate the Reviewer's concerns over the clarity of the data provided. We have decided to only show data on serum triglycerides from mice control, *Gclc* L-KO, *Nrf2* L-KO, and *Gclc/Nrf2* L-DKO mice.

We have included these findings as Figure 5J below and now state:

[figure redacted]

Notably, the repression of lipogenic gene and protein expression and the decreased serum triglyceride levels seen in *Gclc* L-KO mice were reversed in L-DKO mice (Figure 5F-5J and Table S5).

Fig. 1L: List how many independent animals for each genotype are quantified for adipocyte sizes.

We have removed the data pertaining to adipocyte sizes from the manuscript.

Fig. 2E and Fig. 2K, animal numbers for proteomic analyses are not consistent, n=3 or 6?

We appreciate the Review for pointing out this typo. The manuscript legends now state n=6 for animal numbers in Fig. 2E and Fig. 2K.

Fig. 2E, 2K, Fig. 3 A-C: data are not plotted based on Log2 fold change >2. It seems to be FC >2. Please correct.

We appreciate the Review for pointing out this typo. We have corrected Fig. 2E, 2K, Fig. 3 A-C to report “log2 Fold change >1” instead of “Log2 Fold change >2”.

Fig. 4D and Fig. 5H, if there are no differences in survival, no figures are necessary to show as one can hardly differentiate different genotypes.

We appreciate the Reviewers' suggestion and have removed Fig 5H from the manuscript. Fig 4D, now Fig S5C, is maintained but has been edited to show each genotype distinctly.

The number of cohort animals for some quantitative analyses is too low. For example, Fig. S4: n=2 or 4.

We appreciate the Reviewers' suggestions. We have analyzed additional animals to increase the animal numbers.

Animal sex is not provided consistently in most experiments.

We appreciate the Reviewers' concern. We have included animal sex in Figure Legends for panels with sex-dependent findings.

Whether animals were fasted or not before harvesting serum or tissues was not clear. This is important since it affects interpretation of the crosstalk between liver, serum and adipose lipid profiles.

We appreciate the Reviewers' concern. To avoid any confounding factors caused by fasting, animals were not fasted before harvesting serum or tissues.

Fig. 4K include TG levels in WT under normal chow diet.

We appreciate the Reviewers' concern. We have included the TG levels in WT under a normal chow diet in Fig. 4K, now Fig 4J.

Fig. S4K: indicate how many weeks after TAM.

In Fig. S4K, now S7A, we indicate the number of weeks after AAV-TBG-Cre injection.

Reviewer #3 (Remarks to the Author):

This manuscript addresses the effect of profound GSH depletion induced by GCLC deletion on the Nrf2 repression of hepatic triglyceride export leading to lethal weight loss.

1- The condition of such profound sustained hepatic GSH depletion is almost never achieved in any circumstance so not clear what is the relevance of the findings.

We appreciate the Reviewers' concern. However, we believe that sustained hepatic GSH depletion is required to understand the impact of GSH in adult animals. We have provided additional writing and clarification of the relevance of the findings in the Introduction and Discussion.

2- The issue of dissociating GSH depletion and Nrf2 is important and only supported by DKO. Other substantiation is needed – e.g. alternative ways of Nrf2 activation in absence of GSH depletion, e.g. overexpression of Nrf2 or chemical activation of Nrf2. This could be explored in cell culture models.

We appreciate this question from the Reviewer. Several groups have studied Nrf2 activation through disruption of KEAP1 using both genetic and pharmacologic approaches. A Keap1 hypomorph mouse strain was shown to repress the expression of lipid genes and dampen weight gain following a high-fat diet (Slocum SL, Arch Biochem Biophys, 2016, PMID: 26701603). Similarly, a liver-specific Keap1 deletion was shown to lower the expression of lipid genes and reduce liver steatosis induced by a Western diet (Ramadori P, Free Radic Biol Med., 2016, PMID: 26698665). Pharmacologically, KEAP1 activity can be inhibited by barodoxyllone (CDDO). Treatment of mice with CDDO-imidazolide (CDDO-im) lowered the expression of lipid genes and dampened body weight gain caused by a high-fat diet (Shin S, Eur J Pharmacol, 2009, PMID: 19698707). Additionally, the treatment of mice with the NRF2-inducing compound TBE-31 suppressed the expression of lipogenic genes and lowered liver steatosis in mice following a high-fat, high-fructose diet (Sharma RS, Cell Mol Gastroenterol Hepatol, 2017, PMID: 29552625).

We tested how direct NRF2 activation using barodoxyllone-methyl (CDDO-me) compares to NRF2 activation following liver-specific Gclc deletion. Notably, while NRF2 protein accumulation could be observed in whole-protein lysate of liver tissue from liver-specific Gclc KO mice (**Fig. 4E and 5E**), this was barely visible even under high exposure in liver tissue from mice treated with CDDO-me (**Fig. S14A**). Following the enrichment of nuclear lysate, NRF2 protein could be detected in liver tissue from mice treated with CDDO-me (**Fig. S14B**); however, these levels were still not as elevated compared to those observed in liver-specific Gclc deletion mice. We found that treatment with CDDO-me caused induction of NRF2 target genes (i.e., Nqo1), repression of lipogenic enzymes (i.e., Scd1), and lower levels of serum triglycerides (**Figure 5J-5L and Figure S14C-S14E**). In line with the muted accumulation of NRF2 protein, CDDO-me induced NRF2 targets and repressed lipogenic genes to a dramatically lower degree compared to liver-specific Gclc disruption. We hypothesize that we observed a greater induction of NRF2 through liver-specific Gclc deletion (as compared to KEAP1 inhibition with CDDO-me) because GSH prevents NRF2 activation through KEAP1-independent mechanisms. Indeed, it was recently reported that treatment of cells with H₂O₂, which can lower GSH levels, leads to increased expression of NRF2 target genes in a Keap1-independent mechanism (Al-Mubarak BR, Redox Biol, 2021, PMID: 34626892). We have shown the data below and now include the following statement in the manuscript:

[figure redacted]

[figure redacted]

“Our results indicate that GSH supports lipid abundance by preventing NRF2 activation in the liver, as well as through other potential NRF2-independent mechanisms. NRF2 activation has been shown to repress the expression of lipogenic genes¹⁻⁴. NRF2 protein stability is regulated by KEAP1. Pharmacologically, KEAP1 activity can be inhibited by the compound bardoxolone (CDDO)⁵. Therefore, to further dissociate the effects of GSH and NRF2 on lipid abundance and assess if NRF2 activation alone mimics the effects of liver-specific Gclc deletion, we treated Gclc WT mice with CDDO-methyl (CDDO-Me), a form of bardoxolone with increased bioavailability. Treatment with CDDO-Me increased NRF2 expression but not to the level observed in the liver of Gclc L-KO mice, as we could only detect substantial NRF2 expression after enriching the whole protein lysate for a nuclear fraction (Figure S14A-S14B). CDDO-Me treatment increased the expression of NRF2 target genes (Figure S14C-S14D) and lowered the expression of lipogenic genes (Figure 5K, S14E). Consequently, CDDO-Me treatment caused a reduction in serum triglycerides (Figure 5L). However, these resulting effects of NRF2 induction via CDDO treatment were to a dramatically lower degree compared to the effects of NRF2 induction upon GSH depletion in Gclc L-KO mice (Figure 4E-4F and 5D-5E). CDDO-Methyl treatment did not impact body weight, liver damage, or serum glucose and cholesterol levels (Figure S14F-S14J). We hypothesize that we observed a greater induction of NRF2 through liver-specific Gclc deletion (as compared to KEAP1 inhibition with CDDO-me) because GSH prevents NRF2 activation through KEAP1-independent mechanisms. Indeed, it was recently reported that treatment of cells with H₂O₂, which can lower GSH levels, leads to increased expression of NRF2 target genes in a Keap1-independent mechanism⁶.”

3- GSH turnover in the liver is mainly determined by the release into blood and bile leading to maintenance of steady levels of cysteine systemically (Seminars Liver Disease 18:313-329, 1998). Therefore, sustained marked depletion of hepatic GSH will be expected to impair the interorgan homeostasis. What happened to the levels of GSH in extrahepatic organs when hepatic GCLC was deleted?

We appreciate this important question by the Reviewer. We analyzed GSH levels in the kidney, a tissue that is reported to engage in the gamma-glutamyl cycle through GGT1 and support levels of cysteine systemically. Indeed, GGT1 KO mice have lower circulating cystine levels and lower levels of GSH in the kidney¹³. Interestingly, we do not see a decrease in GSH levels in the kidney. This suggests that independent of the liver, other tissues can potentially secrete GSH and maintain systemic cysteine levels.

We have included these findings as Supplementary Figure S5 and now state:

[figure redacted]

No differences in Gclc mRNA were observed in surrounding tissues (Figure S5A), suggesting that the deletion of Gclc was localized to the liver. Further, no differences in GSH levels were found in surrounding tissues (i.e., the kidney) (Figure S5B), suggesting that catabolism liver-derived circulating GSH was not contributing to GSH levels in tissues.

4- Some insight into the extent of GSH depletion and/or Nrf2 activation required to impair triglyceride metabolism in the liver would enhance the manuscript.

We appreciate the suggestion by the Reviewer to examine the extent of GSH depletion and/or Nrf2 activation required to impair triglyceride metabolism in the liver. To test this, we have titrated the amount of AAV-TBG-Cre virus (0.5 , 1 , and 2.5×10^{11} pfu) injected into Gclc^{+/+} and Gclc^{ff} mice to examine the extent of Gclc deletion, GSH depletion, and Nrf2 activation required to impair lipid metabolism. We observed that the injection of an increasingly higher concentration of AAV-TBG-Cre virus caused a greater deletion of Gclc mRNA, reduction of a GSH, induction of NRF2 activity (as measured by increased expression of Nrf2 target gene Nqo1), and repression of lipogenic gene expression. Importantly, we found that a mild induction of NRF2 activity (observed with 0.5×10^{11} pfu) was not sufficient to repress lipogenic gene expression as compared to a strong induction of NRF2 activity (observed with 1 and 2.5×10^{11} pfu). These data are supported by our experiments with the NRF2-inducing compound CDDO-me, which induces a mild increase in NRF2 target gene expression and a mild decrease in lipogenic gene expression.

We have included these findings as Supplementary Figure S8 and now state:

[figure redacted]

Our findings showed that liver-specific *Gclc* deletion results in increased Nrf2 activity, impaired expression of lipogenic enzymes, and decreased circulating triglycerides. However, the extent of GSH depletion required to impact triglyceride metabolism was unknown. To examine this, we treated mice with titrated amounts of the AAV-TBG-Cre virus to induce varying degrees of *Gclc* deletion. Higher viral concentration resulted in greater induction of Nrf2 activity (Figure S8A-S8B). This was associated with a concurrent reduction in lipogenic gene expression and serum triglycerides (Figure S8C-S8E). Particularly, we found the repression of lipogenic genes required a strong induction

of Nrf2 activity, as mild induction of Nrf2 activity (0.5×10^{11} pfu) was not sufficient to repress lipogenic gene expression. Further, we observed higher viral concentration correlated with increased expression of serum markers of liver damage (Figure S8F-S8G). Together, these findings suggest that even minimal levels of GSH in the liver can maintain its ability to support serum triglyceride levels.

References

- 1 Kay, H. Y. *et al.* Nrf2 inhibits LXR α -dependent hepatic lipogenesis by competing with FXR for acetylase binding. *Antioxid Redox Signal* **15**, 2135-2146 (2011). <https://doi.org/10.1089/ars.2010.3834>
- 2 Huang, J., Tabbi-Anneni, I., Gunda, V. & Wang, L. Transcription factor Nrf2 regulates SHP and lipogenic gene expression in hepatic lipid metabolism. *Am J Physiol Gastrointest Liver Physiol* **299**, G1211-1221 (2010). <https://doi.org/10.1152/ajpgi.00322.2010>
- 3 Tanaka, Y. *et al.* NF-E2-related factor 2 inhibits lipid accumulation and oxidative stress in mice fed a high-fat diet. *J Pharmacol Exp Ther* **325**, 655-664 (2008). <https://doi.org/10.1124/jpet.107.135822>
- 4 Kitteringham, N. R. *et al.* Proteomic analysis of Nrf2 deficient transgenic mice reveals cellular defence and lipid metabolism as primary Nrf2-dependent pathways in the liver. *J Proteomics* **73**, 1612-1631 (2010). <https://doi.org/10.1016/j.jprot.2010.03.018>
- 5 Yates, M. S. *et al.* Pharmacodynamic characterization of chemopreventive triterpenoids as exceptionally potent inducers of Nrf2-regulated genes. *Mol Cancer Ther* **6**, 154-162 (2007). <https://doi.org/10.1158/1535-7163.MCT-06-0516>
- 6 Al-Mubarak, B. R. *et al.* Non-canonical Keap1-independent activation of Nrf2 in astrocytes by mild oxidative stress. *Redox Biol* **47**, 102158 (2021). <https://doi.org/10.1016/j.redox.2021.102158>
- 7 Popineau, L. *et al.* Novel Grb14-Mediated Cross Talk between Insulin and p62/Nrf2 Pathways Regulates Liver Lipogenesis and Selective Insulin Resistance. *Mol Cell Biol* **36**, 2168-2181 (2016). <https://doi.org/10.1128/MCB.00170-16>
- 8 Chen, Y. *et al.* Hepatocyte-specific Gclc deletion leads to rapid onset of steatosis with mitochondrial injury and liver failure. *Hepatology* **45**, 1118-1128 (2007). <https://doi.org/10.1002/hep.21635>
- 9 Tian, W., Rojo de la Vega, M., Schmidlin, C. J., Ooi, A. & Zhang, D. D. Kelch-like ECH-associated protein 1 (KEAP1) differentially regulates nuclear factor erythroid-2-related factors 1 and 2 (NRF1 and NRF2). *J Biol Chem* **293**, 2029-2040 (2018). <https://doi.org/10.1074/jbc.RA117.000428>
- 10 Akl, M. G. *et al.* Complementary gene regulation by NRF1 and NRF2 protects against hepatic cholesterol overload. *Cell Rep* **42**, 112399 (2023). <https://doi.org/10.1016/j.celrep.2023.112399>
- 11 Leung, L., Kwong, M., Hou, S., Lee, C. & Chan, J. Y. Deficiency of the Nrf1 and Nrf2 Transcription Factors Results in Early Embryonic Lethality and Severe Oxidative Stress*. *Journal of Biological Chemistry* **278**, 48021-48029 (2003). <https://doi.org/10.1074/jbc.M308439200>
- 12 Chen, L. *et al.* Nrf1 is critical for redox balance and survival of liver cells during development. *Mol Cell Biol* **23**, 4673-4686 (2003). <https://doi.org/10.1128/mcb.23.13.4673-4686.2003>
- 13 Lieberman, M. W. *et al.* Growth retardation and cysteine deficiency in gamma-glutamyl transpeptidase-deficient mice. *Proc. Natl. Acad. Sci. U. S. A.* **93**, 7923-7926 (1996).

REVIEWERS' COMMENTS

Reviewer #1 (Remarks to the Author):

Sufficient amount of efforts have been made by the authors, and most of the questions have been properly addressed.

Reviewer #2 (Remarks to the Author):

The authors did perform extensive experiments to address my main concerns. The manuscript is improved. However, the findings are still incremental. Including the sex-dependent phenotypes in L-DKO mice make the study more confusing without providing clear insights. The following concerns remain:

In Figure 6: the author states: "Importantly, we have examined Gclc-Nrf2 L-DKO at an extended time point of 10 weeks and found that males (but not females) have reduced survival (Fig. 6 and Fig. S15, S16, and S17). This suggests that female mice can compensate for the absence of NRF2 under GSH depletion as compared to males." However, without head-to-head comparison of NRF1 expression in male and female L-DKO mice, I do not think the authors can claim that.

Evidence supporting increased necroptosis in L-KO liver is minimal with only negligible increase in p-MLKL based on IHC (Figure S6A). The caption of Figure S6 does not align with the text which claims no changes of cell death, since there is increased necroptosis in L-KO liver.

Regarding my previous comments: How about the lipogenic gene expression and VLDL secretion in isolated primary hepatocytes from Gclc LKO and DLKO mice? The author stated that "The phenotypes that we describe in the manuscript focus on the impact of liver specific Gclc KO on serum triglycerides in vivo, and we believe that investigating lipogenic gene expression and VLDL secretion in isolated primary hepatocytes is beyond the scope of this manuscript." I do not agree with the authors. Experiments in isolated primary hepatocytes provide direct evidence about whether GSH depletion affects lipogenic gene

expression cell-autonomously. Measurements of VLDL secretion (at least in vivo) will further strengthen the link between liver lipogenic gene expression and serum triglycerides levels, to eliminate the contribution of other tissues in serum TG levels, especially intestine.

The authors claim that “Additionally, we found sex-dependent changes in serum levels of cholesterol and glucose, which might contribute to serum triglycerides (Figure S15E – S15F).” How do you connect serum levels of cholesterol and glucose to triglycerides?

What is the sex of the data presented in Fig. 6B? It does not align with the data in Fig. 6E. In Fig. 6E, male L-DKO mice died about 3-4 weeks post TBG-Cre injection. How can you state that “All mice were sacrificed 10 weeks post AAV-TBG-Cre injection.” How can you obtain Histology in livers from male and female L-DKO mice 10 weeks post TBG-Cre injection? Figure 6C also suggests female L-DKO have 20% mortality. The same concerns exist throughout Figure S15, S16, S17.

Since the authors observed the sex-dependent phenotypes between male and female L-DKO mice after acute and chronic deletion, please include all these information (sex and time post TBG-Cre) in all the figures and texts involving L-DKO mice.

In Figure 5J figure legend: “(J) Relative abundance of serum triglycerides in the liver of WT (n=4), Gclc L-KO (n=4), Nrf2 LKO (n=4) and L-DKO (n=4) mice expressed as Log4 fold change.” It should be Log2 fold change.

Reviewer #4 (Remarks to the Author):

This is a revised paper from Asantewaa et al that examined the role of GSH in lipid abundance in vivo. Using conditional deletion of Gclc in adult mice, they found GSH is essential for lipid abundance. Total body loss of Gclc in adult mice resulted in lower expression of lipogenic enzymes, circulating TGs and fat stores and all died by day 15. However, liver-specific deletion of Gclc in adult mice suffered no mortality at 3 weeks, even

though they also had lower expression of lipogenic enzymes and circulating TGs. They attributed lower TGs to increased NRF2 activation, which represses adipogenesis and proposed normal liver GSH level is a crucial contributor to lipid abundance in vivo by suppressing NRF2. The paper has abundance of data with different genetic deletions, including double KOs of hepatocyte Gclc and Nrf2, as well as activators and inhibitors of the NRF2 signaling pathway. The authors also appeared to have made a great attempt to address comments raised by reviewer #3. However, there remains a number of issues listed below.

Major comments:

1. Novelty remains a big issue. GSH depletion has been shown to protect from liver steatosis, possibly via AMPK activation, and is well known to activate NRF2, which is well documented to repress lipogenic genes.
2. It is odd that liver-specific Gclc KO had no influence on systemic GSH availability, as this contradicts what's known about hepatic GSH in systemic GSH homeostasis. Were the other organs' GSH levels altered in the extended study of the liver-specific Gclc KO?
3. Total body conditional Gclc KOs died by day 15, yet most liver-specific Gclc KOs survived beyond 60 days. Both exhibit similar lipid phenotypes, which supports it is liver GSH that is the major determinant of serum TGs. There is no information provided on the cause of death in the total body KO, other than that the liver, kidney, spleen and pancreas did not appear different from controls histologically.
4. Are the findings applicable under even pathological conditions? A profound sustained hepatic GSH depletion is required to impact on lipid abundance, but this does not occur in human liver diseases and mild depletion had no effect on lipid abundance (Fig. S8).
5. Fig. 4K-M – treatment with GSH-ee – what happened to the liver GSH levels? This should be measured in order to see if the treatment raised GSH level. What is the explanation that while this lowered the Nqo1 mRNA levels, it had no influence on Scd1 mRNA levels or serum TGs?
6. Fig. 5D, F – the rescue on Nqo1 and Scd1 mRNA levels is modest, which means there are NRF2-independent mechanisms. However, serum TG is now normalized (Fig. 5G), which would suggest the effect on serum TG is NRF2-dependent. These findings are contradictory to each other.

7. Authors used a pharmacologic inhibitor of KEAP1 and saw NRF2 induction was not as dramatic as GSH depletion in liver-specific Gclc KO, so they speculate there are KEAP1-independent mechanisms. However, before making that conclusion authors need to see if treatment with CDDO-Me lowered the interaction between NRF2 and KEAP1.

Minor comments:

1. Page 16, discussion – revise this sentence: “We find that the loss of GCLC across all tissue triglycerides and of adult animals results in rapid weight loss and death”.
2. Page 17 – Repression of lipogenic gene expression is NOT completely dependent on NRF2 activation, as shown in Fig. 5F. Discussion should be revised.
3. Page 18 – NAFLD should be changed to the new nomenclature MASLD.

We appreciate the Reviewers' comments and suggestions. The revisions have significantly strengthened the evidence supporting the manuscript's conclusions.

Specific responses to Reviewer comments:

Reviewer comments – Red

Author responses – Black

Text from the manuscript – Bolded in Black

Reviewer #1 (Remarks to the Author):

Sufficient amount of efforts have been made by the authors, and most of the questions have been properly addressed.

We appreciate the positive feedback from Reviewer #1.

Reviewer #2 (Remarks to the Author):

The authors did perform extensive experiments to address my main concerns. The manuscript is improved.

We appreciate the positive feedback from Reviewer #2.

However, the findings are still incremental. Including the sex-dependent phenotypes in L-DKO mice make the study more confusing without providing clear insights.

We respectfully disagree with the opinion of Reviewer #2 that the findings are incremental. Previously, it was suggested that GSH synthesis was required for liver survival (PMID: 17464988). We show that, in contrast to previous findings, GSH synthesis in the adult liver is not required for survival but maintains lipid synthesis by suppressing NRF2. We provide clear mechanistic insight into these phenotypes, showing that 1) Gclc deficiency in the liver lowers serum triglycerides, and this can be rescued with combined deletion of Gclc and Nrf2; 2) the induction of NRF2 upon Gclc deficiency in the liver is specifically due to a loss in GSH levels and not a non-enzymatic function of Gclc; 3) the repression of lipid production caused by Gclc deficiency is dependent on repression of LXR activity; 4) complete loss of Gclc and maximal induction of NRF2 is required to repress lipid production in the liver. Further, we show a sex-dependent requirement for the induction of NRF2 following GSH depletion in the liver, potentially providing clues for sex-dependent etiologies of diseases that involve oxidative stress.

The following concerns remain:

In Figure 6: the author states: "Importantly, we have examined Gclc-Nrf2 L-DKO at an extended time point of 10 weeks and found that males (but not females) have reduced survival (Fig. 6 and Fig. S15, S16, and S17). This suggests that female mice can compensate for the absence of NRF2 under GSH depletion as compared to males." However, without head-to-head comparison of NRF1 expression in male and female L-DKO mice, I do not think the authors can claim that.

We respectfully disagree with the opinion of Reviewer #2. We think the conclusion that "**female mice can compensate for the absence of NRF2 under GSH depletion as compared to males.**" is valid based on the data showing that Gclc-Nrf2 L-DKO males (but not females) have reduced survival. We acknowledge that NRF1 potentially supports survival following the combined

loss of NRF2 and GCLC. We think further research, outside the scope of the current study, is required to fully elucidate the potential compensation by NRF1 in the absence of NRF2 under GSH depletion.

Evidence supporting increased necroptosis in L-KO liver is minimal with only negligible increase in p-MLKL based on IHC (Figure S6A).

We respectfully disagree with the opinion of Reviewer #2. We think that the data provided in Figure S6A clearly show increased staining for p-MLKL on IHC from L-KO liver compared to wildtype liver.

The caption of Figure S6 does not align with the text which claims no changes of cell death, since there is increased necroptosis in L-KO liver.

We appreciate the comment from Reviewer #2. We have revised the caption for Figure S6, which now states, **“Liver Gclc deletion does not cause inflammation, oxidative stress, or apoptosis but increases necroptosis markers. Related to Figure 4.”**

Regarding my previous comments: How about the lipogenic gene expression and VLDL secretion in isolated primary hepatocytes from Gclc LKO and DLKO mice? The author stated that “The phenotypes that we describe in the manuscript focus on the impact of liver specific Gclc KO on serum triglycerides in vivo, and we believe that investigating lipogenic gene expression and VLDL secretion in isolated primary hepatocytes is beyond the scope of this manuscript.” I do not agree with the authors. Experiments in isolated primary hepatocytes provide direct evidence about whether GSH depletion affects lipogenic gene expression cell-autonomously. Measurements of VLDL secretion (at least in vivo) will further strengthen the link between liver lipogenic gene expression and serum triglycerides levels, to eliminate the contribution of other tissues in serum TG levels, especially intestine.

We appreciate the comment from Reviewer #2. While this is interesting, further research outside the scope of the current study is required to investigate the contributions of other tissues (i.e., the intestine) to serum triglyceride levels in liver-specific Gclc KO mice.

The authors claim that “Additionally, we found sex-dependent changes in serum levels of cholesterol and glucose, which might contribute to serum triglycerides (Figure S15E – S15F).” How do you connect serum levels of cholesterol and glucose to triglycerides?

We appreciate the question from Reviewer #2. We have revised the manuscript, which now states, **“Additionally, we found sex-dependent changes in serum levels of cholesterol and glucose (Figure S15E – S15F).”**

What is the sex of the data presented in Fig. 6B? It does not align with the data in Fig. 6E. In Fig. 6E, male L-DKO mice died about 3-4 weeks post TBG-Cre injection. How can you state that “All mice were sacrificed 10 weeks post AAV-TBG-Cre injection.” How can you obtain Histology in livers from male and female L-DKO mice 10 weeks post TBG-Cre injection? Figure 6C also suggests female L-DKO have 20% mortality. The same concerns exist throughout Figure S15, S16, S17.

We appreciate the question from Reviewer #2. Data provided in Figure 6, Figure S15, Figure S16, and Figure S17 are from male and female mice at 10 weeks post AAV-TBG-Cre injection unless humane endpoints were reached earlier. We have revised the caption for Figure 6A to state,

“Mice were sacrificed 10 weeks post AAV-TBG-Cre injection unless humane endpoints were reached earlier.”

Since the authors observed the sex-dependent phenotypes between male and female L-DKO mice after acute and chronic deletion, please include all these information (sex and time post TBG-Cre) in all the figures and texts involving L-DKO mice.

The sex and time post AAV-TBG-Cre injection are provided when applicable.

In Figure 5J figure legend: “(J) Relative abundance of serum triglycerides in the liver of WT (n=4), Gclc L-KO (n=4), Nrf2 LKO (n=4) and L-DKO (n=4) mice expressed as Log4 fold change.” It should be Log2 fold change.

We appreciate the suggestion from Reviewer #2. We have revised Figure 5J to show the data as a Log2 fold change.

Reviewer #4 (Remarks to the Author):

This is a revised paper from Asantewaa et al that examined the role of GSH in lipid abundance in vivo. Using conditional deletion of Gclc in adult mice, they found GSH is essential for lipid abundance. Total body loss of Gclc in adult mice resulted in lower expression of lipogenic enzymes, circulating TGs and fat stores and all died by day 15. However, liver-specific deletion of Gclc in adult mice suffered no mortality at 3 weeks, even though they also had lower expression of lipogenic enzymes and circulating TGs. They attributed lower TGs to increased NRF2 activation, which represses adipogenesis and proposed normal liver GSH level is a crucial contributor to lipid abundance in vivo by suppressing NRF2. The paper has abundance of data with different genetic deletions, including double KOs of hepatocyte Gclc and Nrf2, as well as activators and inhibitors of the NRF2 signaling pathway. The authors also appeared to have made a great attempt to address comments raised by reviewer #3. However, there remains a number of issues listed below.

We appreciate the positive feedback from Reviewer #4.

Major comments:

1. Novelty remains a big issue. GSH depletion has been shown to protect from liver steatosis, possibly via AMPK activation, and is well known to activate NRF2, which is well documented to repress lipogenic genes.

We respectfully disagree with the opinion of Reviewer #4 that novelty remains a big issue. Previously, it was suggested that GSH synthesis was required for liver survival (PMID: 17464988). We show that, in contrast to previous findings, GSH synthesis in the adult liver is not required for survival but maintains lipid synthesis by suppressing NRF2. We provide clear mechanistic insight into these phenotypes, showing that 1) Gclc deficiency in the liver lowers serum triglycerides, and this can be rescued with combined deletion of Gclc and Nrf2; 2) the induction of NRF2 upon Gclc deficiency in the liver is specifically due to a loss in GSH levels and not a non-enzymatic function of Gclc; 3) the repression of lipid production caused by Gclc deficiency is dependent on repression of LXR activity; 4) complete loss of Gclc and maximal induction of NRF2 is required to repress lipid production in the liver. Further, we show a sex-dependent requirement for the induction of NRF2 following GSH depletion in the liver, potentially providing clues for sex-dependent etiologies of diseases that involve oxidative stress.

2. It is odd that liver-specific Gclc KO had no influence on systemic GSH availability, as this contradicts what's known about hepatic GSH in systemic GSH homeostasis. Were the other organs' GSH levels altered in the extended study of the liver-specific Gclc KO?

We appreciate the question from Reviewer #4. It is unclear, however, what previous studies Reviewer #4 is referring to surrounding the impact of hepatic GSH in systemic GSH homeostasis. While this is interesting, further research, outside the scope of the current study, is required to investigate how other organs' GSH levels are altered in liver-specific Gclc KO mice at extended time points (i.e., 10 weeks post AAV-TBG-Cre injection).

3. Total body conditional Gclc KOs died by day 15, yet most liver-specific Gclc KOs survived beyond 60 days. Both exhibit similar lipid phenotypes, which supports it is liver GSH that is the major determinant of serum TGs. There is no information provided on the cause of death in the total body KO, other than that the liver, kidney, spleen and pancreas did not appear different from controls histologically.

We appreciate the question from Reviewer #4. As outlined by Reviewer #4, whole-animal Gclc KO mice rapidly lose body weight and have reduced levels of circulating triglycerides. Liver-specific Gclc KO mice show reduced levels of circulating triglycerides but do not rapidly lose body weight. We hypothesized that the loss of body weight was caused by the loss of Gclc in a tissue other than the liver. Further analysis revealed that whole-animal Gclc KO mice show intestinal damage, including enteritis and colitis. Additionally, we found that intestinal-specific Gclc KO mice rapidly lost body weight and have colitis but do not show reduced levels of circulating triglycerides, suggesting that loss of GSH in the intestine is responsible for the rapid loss of body weight in whole-animal Gclc KO mice. We request that this data be excluded from the manuscript so we can fully explore these phenotypes in future studies.

4. Are the findings applicable under even pathological conditions? A profound sustained hepatic GSH depletion is required to impact on lipid abundance, but this does not occur in human liver diseases and mild depletion had no effect on lipid abundance (Fig. S8).

Reviewer #4 makes an interesting point. While we make no claims about human pathology, we think our findings could extend to human conditions of prolonged GSH depletion, such as chronic liver disease and environmental exposures to toxins. Further work is needed to associate human liver GSH levels during disease states with lipid abundance.

5. Fig. 4K-M – treatment with GSH-ee – what happened to the liver GSH levels? This should be measured in order to see if the treatment raised GSH level. What is the explanation that while this lowered the Nqo1 mRNA levels, it had no influence on Scd1 mRNA levels or serum TGs?

We appreciate the question from Reviewer #4. We found that exceedingly high concentrations of GSH-ee (i.e., 1000 mg/kg) were required to block the induction of NRF2 target genes in liver-specific Gclc KO mice (Figure 4K). We measured GSH levels 24 hours following the final treatment with GSH-ee but found no significant increases in GSH levels in the liver (data not shown). We think the failure of GSH-ee to increase levels of GSH in the liver is due to the poor pharmacodynamics of the delivery of GSH into the tissues of adult animals. Further, we think the treatment with GSH-ee failed to influence Scd1 mRNA levels or serum TGs due to these technical limitations. We have revised the manuscript and now state, **“Treatment of Gclc L-KO with GSH-ee reduced the observed increased Nrf2 target gene expression to basal levels in a dose-dependent manner (Figure 4K and S9A). However, this was not sufficient to rescue the**

decreased lipogenic factors or decreased circulating triglycerides (Figure 4L-4M and S9B), potentially due to the technical limitations associated with delivering GSH to adult animals.”

6. Fig. 5D, F – the rescue on Nqo1 and Scd1 mRNA levels is modest, which means there are NRF2-independent mechanisms. However, serum TG is now normalized (Fig. 5G), which would suggest the effect on serum TG is NRF2-dependent. These findings are contradictory to each other.

We respectfully disagree with Reviewer #4. In Figure 5D, we found that Nqo1 mRNA levels were increased in liver-specific Gclc KO mice by greater than 30-fold compared to WT mice, and this increase was completely rescued in the liver-specific Gclc-Nrf2 KO mice (i.e., L-DKO). We agree with Reviewer #4 that the L-DKO mice did not observe a full rescue in Scd1 mRNA levels, suggesting that, following GSH depletion, NRF2-independent pathways contribute to the control of the transcription of certain lipogenic genes. In the manuscript, we state, “**Nonetheless, the deletion of Nrf2 prevented the increased expression of NRF2 protein and the induction of NRF2 targets upon Gclc deletion in liver tissue (Figure 5D-5E). Notably, the repression of lipogenic gene and protein expression and the decreased serum triglyceride levels seen in Gclc L-KO mice were reversed in L-DKO mice (Figure 5F-5J, and Table S5). For certain lipogenic genes, however, the reversal of downregulated expression was not complete (Figure 5F), suggesting the potential involvement of NRF2-independent pathways. Together, these results indicate that GSH supports lipid abundance by preventing NRF2 activation in the liver.**”

7. Authors used a pharmacologic inhibitor of KEAP1 and saw NRF2 induction was not as dramatic as GSH depletion in liver-specific Gclc KO, so they speculate there are KEAP1-independent mechanisms. However, before making that conclusion authors need to see if treatment with CDDO-Me lowered the interaction between NRF2 and KEAP1.

We appreciate the comment from Reviewer #4. We agree that a separate explanation for the NRF2 induction not being as dramatic as the GSH depletion in liver-specific Gclc KO is that CDDO-me is a weaker inhibitor of NRF2-KEAP1 interaction compared to the ROS generated following GSH depletion. We have revised the manuscript, which now states, “**One hypothesis is that we observed a greater induction of NRF2 through liver-specific Gclc deletion as compared to KEAP1 inhibition with CDDO-me because CDDO-me is a weaker inhibitor of NRF2-KEAP1 interaction compared to the ROS generated following GSH depletion. Alternatively, GSH could have potentially prevented NRF2 activation through both KEAP1-independent mechanisms. Indeed, it was recently reported that treatment of cells with H₂O₂, which can lower GSH levels, leads to increased expression of NRF2 target genes in a KEAP1-independent mechanism⁷⁰. Further research is required to fully elucidate the KEAP1-dependent and -independent mechanisms of GSH-dependent regulation of NRF2.**”

Minor comments:

1. Page 16, discussion – revise this sentence: “We find that the loss of GCLC across all tissue triglycerides and of adult animals results in rapid weight loss and death”.

We appreciate Reviewer #4 identifying this typo. We have revised the manuscript, which now states, “**We find that the loss of GCLC across all tissues of adult animals results in rapid weight loss and death.**”

2. Page 17 – Repression of lipogenic gene expression is NOT completely dependent on NRF2 activation, as shown in Fig. 5F. Discussion should be revised.

We appreciate the comment from Reviewer #4. In the manuscript, we state, “**Notably, the repression of lipogenic gene and protein expression and the decreased serum triglyceride levels seen in Gclc L-KO mice were reversed in L-DKO mice (Figure 5F-5J, and Table S5). For certain lipogenic genes, however, the reversal of downregulated expression was not complete (Figure 5F), suggesting the potential involvement of NRF2-independent pathways.**”

3. Page 18 – NAFLD should be changed to the new nomenclature MASLD.

We appreciate Reviewer #4 identifying this typo. We have revised the manuscript, which now states, “**In mouse and rat models of choline and methionine-deficient diet-induced metabolic dysfunction-associated steatotic liver disease (MASLD), females are reported to be more protected than their male counterparts⁹⁰.**”